



# Upgraded global mapping information for earth system modelling: an application to surface water depth at ECMWF

Margarita Choulga[1], Ekaterina Kourzeneva[2], Gianpaolo Balsamo[1], Souhail Boussetta[1], Nils Wedi[1]

[1]Research Department, European Centre for Medium-range Weather Forecasts (ECMWF), Reading, RG2 9AX, UK
5 [2]Research Department, Finnish Meteorological Institute (FMI), Helsinki, FI-00560, Finland

*Correspondence to*: Margarita Choulga (margarita.choulga@ecmwf.int)

**Abstract.** Water bodies influence local weather and climate, especially in lake-rich areas (e.g. lake regions in Canada and northern Russia). In 2015 a parametrization to represent inland water bodies was included in the Integrated Forecasting System (IFS) used operationally at the European Centre for Medium-Range Weather Forecasts (ECMWF) to produce global weather 10 predictions. The parameterization is based on the Fresh-water Lake model (FLake). In the IFS, FLake runs on any inland surface grid-point. It thus depends on global, realistic and complete lake depth and lake cover data as input. Operationally used lake depths make use of the Global Lake Data Base (GLDBv1) over lakes, a default value of 25 m where lake depth information is missing, and over the ocean bathymetry from ETOPO1. In this study we present the upgraded GLDBv3 dataset, which makes use of mean depth estimates based on geological origin over land, and the GEBCO ocean bathymetry. To assess the 15 impact of using GLDBv3 instead of GLDBv1 in-situ measurements of lake water surface temperatures and information on ice formation/disappearance dates for 27 lakes collected by the Finnish Environment Institute were used for verification. The dataset includes daily temperature measurements and ice formation/disappearance dates recorded by observers. A set of surface experiments were carried out using the IFS and atmospheric forcing from the ERA5 reanalysis to test the operational and new lake depths. These simulations were done at a grid spacing of about 9 km and covered the 5-year period 2010–2014. When 20 verified against in-situ lake depth measurements, the new lake depths in GLDBv3 have a much lower mean absolute error, bias and standard deviation error compared to the current operationally used GLDBv1 depths. Indirect verification was used to compare measured and modelled lake water surface temperatures and ice formation/disappearance dates. On average the mean absolute error of lake surface temperatures is reduced by 13.4 %, biases are reduced by 12.5 % and the standard deviation error by 20.3 % when GLDBv3 is used instead of GLDBv1. Seasonal verification of mixed layer depth temperature and ice 25 formation/disappearance dates (using lake mixing seasons) revealed a cold bias in meteorological forcing from ERA5. However, for spring, summer and autumn verification confirms an overall reduction in the errors in surface water temperatures. For winter verification based on ice formation/disappearance dates shows no statistically significant change in ice disappearance date errors.





# 1 Introduction

A lake can be defined as a significant volume of water, which occupies a depression in the land and has no direct connection with the sea. Inland water bodies are often referred to as lakes when the lateral movement of the water is negligible, and as rivers when there is a sizeable lateral transport, although a clear separation is often complex and varies in time. Despite these complexities, in the following we use the term lakes in the broad sense of any inland water body which has negligible lateral movement of water. Globally lakes occupy about 3.7 % of the land surface (Borre, 2014; Verpoorter et al., 2014). According to the latest calculations the total number of lakes with a water surface area not less than 0.002 $km^2$, is 117 million (excluding Greenland and Antarctica), and their combined area is about 5 million $km^2$ (excluding the Caspian Sea) (Borre, 2014; Verpoorter et al., 2014). Lakes are distributed very unevenly. Most lakes are situated in Boreal and Arctic climate zones 45-75 °N (Borre, 2014), namely in Canada, the Scandinavian Peninsula, Finland and Northern Russia and Siberia. Lakes influence local weather conditions and local climate. For example, during freezing and melting the lake surface radiative and conductive properties and the latent and sensible heat released to the atmosphere changes dramatically, resulting in a completely different surface energy balance (Eerola et al., 2010; Mironov et al., 2010a; Samuelsson et al., 2010; Rontu et al., 2012). Lake Ladoga in Russia can generate low clouds, which lead to an increase in 2-meter temperatures of up to 10 °C in neighbouring Finland (Eerola et at., 2014). The Great Lakes in the USA intensify winter snow storms (Hjelmfelt, 1990; Notaro et al., 2013; Vavrus et al., 2013). During summer in the Boreal zone lakes usually cause a decrease in the amount of precipitation (Samuelsson et al., 2010). The African Lake Victoria generates night convection with intensive thunderstorms, which leads to the death of thousands of fisherman every year (Thiery et al., 2015; Thiery et al., 2017). Lakes can also influence global climate by affecting the carbon cycle (Tranvik et al., 2009) and methane emissions (Stepanenko et al., 2016). Small shallow thermokarst lakes located at Boreal and Arctic latitudes in the permafrost thaw area are rich in nutrients, which affect the $CO_2$ budget (Walter et al., 2006; Walter et al., 2007; Stepanenko et al., 2012). This type of lake is most common (representing approximately 77 % of the lakes globally), and in general has a small surface area (0.002-0.01 $km^2$) and a big surface-to-volume ratio. This shape characteristics are important as carbon dioxide and methane degassing takes place through the lake's surface (Borre, 2014; Verpoorter et al., 2014).

The effect of lakes is handled in Numerical Weather Prediction (NWP) and climate models through parametrization, which needs information on the locations of the lakes and their morphological characteristics. However, their representation within global models may be problematic because 90 million of the world's lakes range between 0.002 to 0.01 $km^2$ in size (Borre, 2014; Verpoorter et al., 2014). To date, the majority of the morphological parameters of these lakes have not been measured, not to mention constantly monitored! Reasons for this include: (i) most of these lakes are too small and common to have specially dedicated measuring campaigns, or (ii) they are situated in very remote and hard to reach areas. In NWP lakes with areas smaller than the model grid-box size are considered to be sub-grid features. For example, the high-resolution version of the Integrated Forecasting System (IFS) model at European Centre for Medium-range Weather Forecasts (ECMWF) uses a grid spacing of approximately 9 km. In this configuration lakes with a surface area of less than 81 $km^2$ are considered to be





sub-grid. The effect of both sub-grid and resolved lakes in NWP and climate modelling is taken into account through parameterization. However, to represent the sub-grid lakes, the lake fraction (relative to the model grid size) is needed.

At ECMWF, the lake parameterization was introduced in 2015 by including the Fresh-water Lake model FLake into the IFS (Mironov et al., 2006; Mironov, 2008; Mironov et al., 2010b; Mironov et al., 2012). To represent surface heterogeneity, the

Tiled ECMWF Scheme for Surface Exchanges over Land incorporating land surface hydrology (HTESSEL) was used. This computes surface turbulent fluxes (of heat, moisture and momentum) and skin temperature over different tiles (vegetation, bare soil, snow, interception and water) and then calculates an area-weighted average for the grid-box to couple with the atmosphere (Balsamo et al., 2012; IFS Documentation, 2017). A new tile, representing lakes, reservoirs, rivers and coastal waters, was introduced (Dutra et al., 2010; IFS Documentation, 2017; see http://www.flake.igb-berlin.de/papers.shtml) in

HTESSEL based on the FLake model. Currently FLake only accurately represents fresh-water lakes, but in the future its large research community plans to also include representation of saline water. FLake is a one-dimensional model, which uses an assumed shape for the lake temperature profile including the mixed layer (uniform distribution of temperature) and the thermocline (its upper boundary located at the mixed layer bottom, and the lower boundary at the lake bottom). The model also contains an ice module, a snow module and a bottom sediments module. At present FLake runs in IFS with no bottom

sediment and snow modules (snow accumulation over ice is not allowed and snow parameters are used only for albedo purposes). In the implementation in IFS lake ice can be fractional within a grid-box with inland water (10 cm of ice means 100 % of a grid-box or tile is covered with ice; 0 cm of ice means 100 % of the grid-box is covered by water; in between a linear interpolation is applied) (Manrique-Sunen et al., 2013). At present, the water balance equation is not included for lakes and the lake depth and surface area are kept constant in time (IFS Documentation, 2017). FLake also requires the lake fraction,

*Fr_lake*, and lake depth (preferably bathymetry), *D_water*, and lake initial conditions. *D_water* is the most important external parameter that FLake uses. Note that the IFS model is a global spectral NWP model, which uses different setups for its climate, ocean and ensemble run calculations and different horizontal resolutions. Currently, the highest operational resolution is 9 km (Tco1279; the resolution of the IFS is indicated by specifying the spectral truncation prefixed by the acronym Tco for triangular-cubic-octahedral). It is important, that lake parametrization is consistent with other external model parameters on

different resolution grids.

Under the framework of the continuous upgrade of the ECMWF IFS model lake related data are updated. The implementation of updates should be straightforward with a minimum disturbance to forecast production. Attention should be paid to coastal waters and areas with changes to inland water bodies, and significant depth changes to large lakes. The *D_water* field should be updated with the latest available information to ensure that depths are close to observed values, as overestimated depths can

be blamed for cold biases in summer temperatures or lack of ice. A realistic bathymetry can be obtained from new in-situ measurements and high-resolution datasets, and a re-evaluation of the default depths.

The aim of this research is to improve forecasts of surface parameters in ECMWF's IFS model by upgrading the lake model *Fr_lake* and *D_water* fields with newly available information. This includes providing consistency between lake data and other



land surface fields. The impact of these innovations was studied. This new algorithm for modelling inland water and ocean water separation may be used by anyone in the environmental modelling community.

The paper is organised as follows. Section 2 describes the "Data" and includes the description of the physiographic datasets used to generate the lake parameters. Section 3 discusses the "Methods" applied to the datasets, both for the currently

operational and upgraded versions. Verification of IFS model experiments against in-situ measurements of lake depth, lake surface water temperature and ice formation/disappearance dates, and a discussion of the results and further developments are covered in Section 4 on "Verification and discussion". The main results, a discussion and further research guidance are covered in the "Conclusion" in Section 5.

## 2 Data

The physiographic datasets used in the IFS model to generate the lake parameters are described here, both for the current and upgraded versions. In addition, descriptions of the other lake related land surface parameter datasets are described. Firstly, *Fr_lake* is related to land use. There are a lot of regional and global ecosystem datasets such as Corine (CLC2006 technical guidelines, 2007) and Ecoclimap (Champeaux et al., 2004), that provide information on land cover types, including inland water (lakes, rivers, etc.).

For land cover types, ECMWF uses the global map GlobCover 2009 (Bontemps et al, 2011; Arino et al., 2012) which has a nominal resolution of 300 meters. This land cover map is used by many limited area models (e.g. COSMO), and has been proven to be an accurate and reliable source of data for NWP modelling (Arino et al., 2012; Quaife and Cripps, 2016). GlobCover 2009 is derived from an automatic, regionally-tuned classification of a time series of global Medium Resolution Imaging Spectrometer Instrument Fine Resolution (MERIS FR) mosaics for the year 2009. It consists of a global land cover

map on a Plate-Carree (WGS84 ellipsoid) projection covering the Earth. Its legend is compatible with the GLC2000 (Bartholome and Belward, 2005) global land cover classification and accounts for 22 land cover classes defined with the United Nations (UN) Land Cover Classification System (LCCS). A 23rd class (coded as "230") has been added to the final legend to account for pixel with no data (Bontemps et al, 2011). GlobCover 2009 land cover map is available from 60 °S to 85 °N, but contains only one "water" cover type, and hence does not distinguish between ocean (sea) and inland water bodies

(lakes, rivers, etc.).

Over polar regions, for the land cover map ECMWF uses the high-resolution Radarsat Antarctic Mapping Project (RAMP) Digital Elevation Model (DEM) Version 2 (RAMP2) data (Liu et al., 2015) for Antarctica. These data are on a 1 km (30") grid in Polar Stereographic coordinates (IFS Documentation, 2017) and are provided as raw binary (the only values 0 = water and 1 = land). In the Arctic, north of 85 °N no land is assumed.

For the upgrade of lake location in selected places, Digital map database of Iceland and Global Surface Water Explorer data are used. National Land Survey of Iceland are constantly reviewing and processing the Digital map database of Iceland (IS 50V). It is based on a variety of sources and data such as GPS-tracking for roads, aerial photographs, SPOT-5 satellite images





and data from other agencies and municipalities. IS 50V consists of 8 layers, including hydrology and coastline. Layers are presented in conical Lambert projection (reference is ISN93 or ISN2004). For our purposes, only coastline and hydrology layers are used to update water distribution for Iceland, these were processed by the Icelandic Meteorological Office (Bolli Palmason and Ragnar Heiðar Þrastarson, personal communication 2018).

The Joint Research Centre (JRC) has created a 30-meter (1") horizontal resolution Global Surface Water Explorer (GSWE) dataset by using Landsat 5, 7 and 8 individual full-resolution 185 km$^2$ global reference system II satellite images over the past 32 years (between March 1984 and October 2015) to map the spatial and temporal variability of global surface water and its long-term changes. These satellites have a near polar orbit, and provide global coverage every 16 days (the individual satellite orbits are such that when two operate concurrently there is an eight-day revisit period). Thermal imagery and the contrasting

spectral properties of water and other features (including snow, clouds, shadows, bare rock and vegetated land) in the Landsat sensors' six visible, near and shortwave infrared channels were used within the expert system to separate pixels acquired over open water from those acquired over other surfaces. Validation of the system shows less than 1 % of false water detections and less than 5 % of missed water surfaces out of 40'000 control points from around the world and during the 32 years (Pekel et al., 2016). GSWE consists of several datasets that show different facets of surface water dynamics. For IFS lake information

upgrade, the Water Transitions facet is used, which shows changes in water classes between the first and last years in which reliable observations were obtained. These are the following:

(0) No water – water was not detected in this place,

(1) Permanent – unchanging permanent water surfaces,

(2) New Permanent – conversion of a no water place into a permanent water place,

(3) Lost Permanent – conversion of a permanent water place into a no water place,

(4) Seasonal – unchanging seasonal water surfaces,

(5) New Seasonal – conversion of a no water place into a seasonal water place,

(6) Lost Seasonal – conversion of a seasonal water place into a no water place,

(7) Seasonal to Permanent – conversion of seasonal water into permanent water,

(8) Permanent to Seasonal – conversion of permanent water into seasonal water,

(9) Ephemeral Permanent – no water places replaced by permanent water that subsequently disappeared within the observation period,

(10) Ephemeral Seasonal – no water places replaced by seasonal water that subsequently disappeared within the observation period,

(255) No data – no reliable observations were obtained.

This map is used to upgrade only certain geographical regions (i.e. Australia, Aral Sea, Alqueva Reservoir).

The lake depth is specified according to Global Lake DataBase, v1 and v3, (Kourzeneva, 2010) and (Choulga et al., 2014), for operational and upgraded versions respectively. In 2008 GLDBv1 was developed for implementation in lake parameterization schemes in NWP and climate modelling (Kourzeneva, 2010). GLDBv1 uses:



(i) the mean depth for individual lakes (~ 13'000 lakes) from different regional databases,

(ii) the global lake mask created from Ecoclimap2 ecosystem dataset (Champeaux et al., 2004), and

(iii) bathymetry data for 36 large lakes from ETOPO1 (Amante and Eakins, 2009) and digitized navigation and topographic maps.

To combine individual lake depth data with a raster cover map, an automatic probabilistic mapping method is used, see (Kourzeneva et al., 2012) for more information. The result was a global lake depth data set on a 30" (~ 1 km) grid. When there was a lake on the map, but its depth value was unknown from the individual lake dataset, the "default" depth of 10 m was used. GLDBv1 is used in the IFS operational setup. In GLDB later versions, the "default" depth was the main subject of study. GLDBv1 was upgraded with indirect mean depth estimates, depending on the geological origin of lake. The geological

approach, used for the depth estimation of uninspected freshwater lakes, assumes that water bodies of the same origin and the same age should have similar morphological parameters, see (Choulga et al., 2014) for more information. An innovative algorithm, which combined information about lake location and morphological parameters, and surface geological and tectonic information was developed and applied. Globally 374 regions (141 for boreal climate zone and 233 for the rest of the globe) with a homogeneous geological origin of lakes were outlined. The typical lake depth values were derived from (1) the

individual lake dataset and global gridded lake depth map statistics, (2) expert judgment, and (3) lists with different lake types, exceptional for the region with the same lake origin. The recent version of the dataset is GLDBv3. Its main differences from GLDBv1:

(i) increase of the individual lake list by ~ 1'500 lakes,

(ii) addition of extra bathymetry data for all navigable and most of non-navigable Finish lakes,

(iii) addition of indirect mean depth estimates based on lake geological origin,

(iv) use of the derived analytical equations to define the lake mean depth from the lakes' area and boreal zones climate type,

(v) introduction of freshwater/saline lake differentiation: the "default" depth for freshwater lakes is set to 10 m, for saline lakes 5 m,

(vi) introduction of two lists with exceptions: artificial lakes (reservoirs) with unknown depths and crater (caldera) lakes with

the "default" depths of 10 and 50 m respectively.

Verification of indirect depth estimates (based on geological origin) against new observations for 353 Finish lakes showed 52 % bias reduction (from 5.4 m in GLDBv1 to 2.6 m in GLDBv3) and 34 % RMSE reduction (from 6.1 m in GLDBv1 to 4.0 m in GLDBv3); improvements in the depth estimates are proved to be statistically significant. In this study GLDBv3 is used to upgrade the IFS lake information.

Operationally, the Caspian Sea bathymetry is from ~ 4 km resolution digitalized data (Cavaleri, personal communication 2008); the Great Lakes, the Azov Sea and the ocean – bathymetry from Global Relief Model ETOPO1 (Amante and Eakins, 2009) with the horizontal resolution 1' (~ 2 km). ETOPO1 consists of regional and global datasets, and bathymetry estimates from satellite altimetry for unsurveyed ocean areas. Horizontal and vertical datum of the model are WGS 84 geographic and "sea level" accordingly.



The upgraded bathymetry for the Caspian Sea, the Azov Sea and the ocean is from General Bathymetric Chart of the Oceans (GEBCO) (Weatherall et al., 2015). Published in 2014, GEBCO is a global terrain model for ocean and land with a 30" (~ 1 km) global grid of elevations. It is largely generated by combining new versions of regional bathymetric compilations from the International Bathymetric Chart of the Arctic Ocean, the International Bathymetric Chart of the Southern Ocean, the Baltic Sea Bathymetry Database, and data from the European Marine Observation and Data network bathymetry portal, quality-controlled ship depth soundings with interpolation between sounding points guided by satellite-derived gravity data. The dataset is accompanied by auxiliary data, where each cells value is identified as based on actual depth values or predicted ones.

## 3 Methods

### 3.1 Current status

The IFS is a global model, and according to its design lake parameterization runs on each surface grid-point, whether the simulation results in this point are used later or not. Independently on the resolution, missing values are not allowed to ease the interoperability of the output at diverse spatial resolutions of IFS model.

Main physiographic fields that govern use of all land-surface parameterization results in the IFS are the land fraction ($Fr\_land$) and corresponding land-water binary mask ($LWM$, 0 = water and 1 = land). $Fr\_land$ provides information about land and water (oceans, seas, lakes, rivers, etc.) fraction in each model grid-box of the underlying surface. In the IFS, model grid-box is land dominated if more than 50 % of the actual surface is land (Manrique-Sunen et al., 2013) (i.e. $Fr\_land > 50\ \% \rightarrow LWM = 1$). All sub-grid water in the land-dominating case is treated as lake water (simulated by FLake). If a grid-box is water dominated (i.e. $Fr\_land \leq 50\ \% \rightarrow LWM = 0$), then extra knowledge of water type is required, as salt ocean and dominantly freshwater lakes and rivers have different physical properties and are treated with different model parameterizations. Both $Fr\_land$ and $LWM$ are grid-dependent. Primarily, $Fr\_land$ is calculated from the land-cover maps (operationally from GlobCover 2009 and RAMP2) by aggregating the "land"-type information on a certain grid. Then $LWM$ is produced. Note that since GlobCover 2009 does not distinguish between ocean (sea) and inland water, $LWM$ also do not distinguish between them.

To distinguish between ocean and inland water, a binary lake mask ($LKM$, 0 = non-lake and 1 = lake) is produced from $LWM$ using a flood-filling algorithm for different resolutions and grids. The idea of this algorithm is to start from a seed somewhere in the open ocean on $LWM$ and let the flood-filling procedure (IFS Documentation, 2017) march through all connected water points (i.e. where $LWM = 0$) marking them as non-lake (i.e. with $LKM = 0$); unmarked points with $LWM = 0$ are not connected to the ocean and stand for the inland water bodies (i.e. $LKM = 1$). The reasons for applying this method instead of using $LKM$ produced from external sources (e.g., from GLDBv1) are the following. Various sources of information almost always have some compatibility errors, in this case – spatial distribution errors – inland water bodies from different inventories can have variations in location, shape and size. It is vital to have $LKM$ consistent with $LWM$, otherwise ocean water can surprisingly appear on the Tibetan Plato. Also, a new high resolution updated $LWM$ appear much earlier than $LKM$ based on them, and usually with lower resolution. As in NWP the quality (accuracy and reliability) of water land data is extremely important,





having an up-to-date high resolution *LWM* is very appealing. This leads to necessity of an in-house algorithm to generate *LKM* from the chosen *LWM* dataset. Issues here are grid-dependency and low accuracy. Some lakes are very close to the sea, and especially for low resolutions, the flood-filling algorithm just fills them up as ocean. Issue was resolved by manually blocking coastal lakes. Another issue was that some narrow parts of the ocean (e.g. fjords in Norway and Greenland) were not filled up

by the flood-filling algorithm (leaving them to freeze as freshwater bodies). Solution here was to use a latitude-dependent threshold for *LWM* (to distinguish water from land) while using the flood-filling algorithm, with lower values in mid- and low-latitudes, and higher values at high latitudes (IFS Documentation, 2017). Finally, FLake results are used for the grid-boxes with $LWM = 1$ or with $LWM = 0$ & $LKM = 1$, using $Fr\_lake = 1 – Fr\_land$. This algorithm is applied separately for each IFS grid with different horizontal resolutions (operational (~ 9 km, Tco1279), climate, ocean, and ensemble).

Since FLake runs in each grid box independently on $Fr\_lake$, $D\_water$ field should be global, even if $D\_water$ values for some points are dummy. To obtain the global depth field with the ocean/lake depth in each grid-box and no missing values, the following steps were made: (1) data from GLDBv1 with 1 km native resolution were aggregated to a 5' grid, (2) in all inland points where GLDBv1 has no information a default value of 25 m is assumed, (3) the minimum depth value is set to 2 m; the Great lakes, the Azov Sea and the Caspian Sea are treated as lakes with (4) the Caspian Sea bathymetry from ~ 4 km resolution

digitalized data (Cavaleri, personal communication 2008), and (5) the Great lakes, the Azov Sea and the ocean bathymetry are from ETOPO1 (Balsamo et al., 2012; IFS Documentation, 2017). Finally, the resulting field is interpolated on various IFS grids and resolutions.

Main disadvantage of the current ocean/inland water separating procedure is simplification of a complex coastline (e.g. Finland, Norway) and neglect of small islands. On coarser resolution narrow land parts that separate freshwater lakes and

saline ocean disappear (land fraction becomes too small) and coastal lakes and wide estuaries are treated as ocean (the surface temperature is extrapolated from Sea Surface Temperature of the nearest ocean grid point), which can lead to no ice conditions during winter in high latitudes or rather low temperatures and almost no diurnal cycle during summer. One example is disappearing islands that separate freshwater lake Alexandrina in South Australia from saline Great Australian Bight (Indian Ocean), which results into flooding of the freshwater lake with saline ocean, and in modelling perspective to the completely

different surface temperature. Figures 1 and 2 left columns show results of operational $Fr\_land$ and $Fr\_lake$ fields combination (remaining fractional ocean part) at 9 km (Tco1279, upper plots) and 32 km (Tco319, lower plots) horizontal resolutions over Finland and North-Western Russia (59-72 °N, 20-42 °E) and North-Eastern Russia (60-74 °N, 122-163 °E) regions respectively. These plots show how use of the current ocean/inland water separating procedure leads to deep ocean penetration into land and/or separated ocean parts over the land at coarser resolutions. For example, Fig. 1 left column upper plot at 9 km

resolution shows neat separation of inland water and ocean, and Fig. 1 left column lower plot at 32 km resolution shows that same water separation procedure leads to deep ocean penetration inlands filling lake Saimaa with salt water through pixel, that became not land dominated on coarser resolution. In addition, several inaccuracies were reported in inland water distribution, such as too wet Australia and omission of Alqueva Reservoir – the biggest man-made lake in Western Europe. All these features required an urgent update.


## 3.2 Updates

The new way of creating lake fields is first to create *LKM* compatible with *LWM* at 1 km resolution regular latitude longitude grid, then to interpolate both to needed resolution and grid. This will allow to preserve water fractions of both types at any resolution independently from *Fr_land*. Figures 1 and 2 right columns give quick peek on *Fr_land* and *Fr_lake* fields

combination (remaining fractional ocean part) created with the new way at 9 km (Tco1279, upper plots) and 32 km (Tco319, lower plots) horizontal resolutions over Finland and North-Western Russia (59-72 °N, 20-42 °E) and North-Eastern Russia (60-74 °N, 122-163 °E) regions respectively. These plots show how use of the new ocean/inland water separating procedure prevents deep ocean penetration into land and/or separation of ocean parts over the land at coarser resolutions. Proposed methodology is designed bearing in mind quite prompt update of global ecosystem maps: new satellite-based products become

freely available with higher and higher resolution more often. To ease the *LKM* compatible with *LWM* upgrade process, water type separation procedure is as automated as possible. *D_water* is the main parameter to drive lake parameterization. In IFS surface scheme FLake runs on each grid-point independently from the *Fr_lake*, so *D_water* field should be global and as realistic as possible. To achieve this, newer dataset versions, various data source compilation and innovative approaches were used.

The new way of generating *LKM* field was (1) to start with 1 km *LWM* and (2) to create a consistent 1 km *LKM*, then (3) to convert binary *LKM* field into fractional *Fr_lake* field, and finally (4) to interpolate it to all IFS grids and resolutions. In this case separation between ocean and inland water is done only once at rather high horizontal resolution (~ 1 km), which still preserves a lot of coastal features, but is computationally (and in data handling sense) cheaper, than the nominal resolution of GlobCover 2009 or GSWE (~ 300 meters and ~ 30 meters respectively).

First step was to aggregate the water cover from initial GlobCover 2009 10" map to 30" (43200/21600 grid-boxes along longitude/latitude) horizontal resolution. At the end of this step aggregated *LWM* was also corrected at certain regions where big water distribution errors were reported. Regions and sources are following.

The Aral Sea is an endorheic lake that used to be one of the four largest lakes in the world. In 1960 its water surface area was 68'900 km$^2$. However, the Aral Sea is shrinking. According to historical records this process started at least in the middle of

25 the 18$^{th}$ century and was accelerated in 1960's after massive diversion of water for cotton and rice cultivation. GlobCover 2009 shows the Aral Sea for 1998 when its water surface area was 28'990 km$^2$ (less than a half of its initial size) (Duhovny et al., 2017), see Fig. 3 upper left plot. Nevertheless after 1998 shrinking continued. The Aral Sea water surface area started stabilizing only in 2014 at area 7'660 km$^2$ (almost 9 times less of its initial size), due to the major Aral Sea recovery program launched in 2001 by the president of Kazakhstan and supported by the World Bank (ENS, 2008), see Fig. 3 upper right plot.

On the updated map, an up-to-date Aral Sea water distribution from GSWE replaced an outdated one from GlobCover 2009. Only currently present water types were used, i.e. Permanent, New Permanent and Seasonal to Permanent.

The Alqueva reservoir is the largest man-made water body in western Europe, and it is completely omitted on GlobCover 2009, see Fig. 3 lower left plot. Its surface area is ~ 210 km$^2$ with minor inter-/annual variability (Miguel Potes and Rui





Salgado, personal communication 2017). An up-to-date Alqueva reservoir water distribution from GSWE based on Permanent, New Permanent and Seasonal to Permanent water types replaced one from GlobCover 2009, see Fig. 3 lower right plot.

Australia is the 6[th] largest (by total area) country in the world with vast number of lakes. Lakes are predominantly dry and salt, located in the flat desert regions. Excess inland water on GlobCover 2009 map was reported for the South-East part of Western

Australia and North part of South Australia (20-30 °S, 130-140 °E), as illustrated by Fig. 4. The left plot shows region in question on GlobCover 2009, with shallow endorheic Kati Thanda-Lake Eyre (28.37 °S, 137.37 °E) in its lower right corner. This lake fills on rare occasions, only few times a century. Here it is seen in its maximum extent. Right three plots show the same region on GSWE Water Transitions map with different water class combinations. The combination of permanent, new permanent and seasonal to permanent water classes reflects permanent water, see second from the left plot. This combination

has almost no inland water, except artificial lake Moondarra (20.59 °S, 139.54 °E) and lake Machattie area (24.90 °S, 139.50 °E), which consists of three lakes: Mipia (usually retains water until the following flood season), Koolivoo (usually dries up by early summer) and Machattie (flooded ~ once in three years). Lakes in lake Machattie area are fresh when filled by floods but become saline as they dry out. If seasonal, new seasonal and permanent to seasonal water classes (which reflect seasonal water) are added, see third from the left plot, then region in question has more water, yet much less than on GlobCover 2009.

If also ephemeral permanent and ephemeral seasonal water classes (which stay for ephemeral water) are added, see right plot, region in question gets even more water than on GlobCover 2009, which was reported as too wet! To make a choice of all year-round plausible water distribution for Australia, experts from Australian National University and Bureau of Meteorology were consulted. It was explained that there are large scale ephemeral inundations in inland Australia, but most of them are occasional rather than seasonal (Albert van Dijk, personal communication 2017). Based on this, it was decided to use the

combination of permanent, new permanent and seasonal to permanent water classes from GSWE Water Transitions map as a whole year static water distribution for Australia, see Fig. 4 second from the left plot. This corresponds well with Water Observations from Space for Australia (see http://www.ga.gov.au/interactive-maps/#/theme/water/map/wofs).

Iceland is located around 63-67 °N which makes it quite poor for reliable satellite observations, also due to much cloud and cloud shadow conditions. Figure 5 left plot shows the GlobCover 2009 water distribution for Iceland. If possible, it is good to

compliment these data with ground observations (e.g. theodolite, lidar). Here, Digital map database of Iceland provided by Icelandic Meteorological Office and referred as the best available for the region source of water distribution information (Bolli Palmason and Ragnar Heiðar Þrastarson, personal communication 2018) was used, see Fig. 5 right plot.

Then corrected *LWM* produced from GlobCover 2009 (which is available from 85 °N to 60 °S) was combined with RAMP2 dataset over Antarctica and assumption of no land north of 85 °N. The resulting field is an updated *LWM*, further used for

upgraded *LKM* creation.

Next step is dividing of *LWM* water into inland and ocean parts. At the beginning the basic flood-filling algorithm was used. However, with the fine ~ 1 km resolution problematic regions with the deep ocean into land penetration (through river estuaries) or merge of different inland water bodies were revealed. Figure 6 shows results of water separation with different techniques at several geographical locations. Upper row plots display no ocean/inland water separation, and middle row plots





– separation with basic flood-filling algorithm. Left column plots show region of Finland and North-Western part of Russia, where inland water is neatly separated by the basic flood-filling algorithm. Middle column plots show region of St Lawrence River with light ocean penetration into land through the St Lawrence River and its lakes (i.e. Saint Pierre, Saint Louis and Two Mountains) and the Ottawa River. Right column plots show region of the Amazon River with deep ocean penetration into the

land through the estuary of the Amazon River and nearby lakes (e.g. Grande do Curuai, Itarim, etc.), as well as the estuaries of the Xingu River and the Tocantins River with the Tucurui Reservoir. This Amazon River region example also shows several inner waterbodies merge, which makes it extremely challenging to automatically map individual lake depth with each water body, as was done in (Kourzeneva et al., 2012) for mapping lake depths for GLDB.

Specially for these complicated situations, when separation should be based on rather physical and geographical then

geometrical features, the innovative waterbody separation algorithm was developed and applied. In general, the algorithm allows to separate narrow rivers or bays from large water bodies (e.g. lakes or seas). Since it is based on something more than just geometry, it contains 2 parameters, which depend on the resolution and complexity of the regions' coastline. These parameters should be defined beforehand relying on the expert opinion (i.e. tuning parameters). The algorithm is pixel-by-pixel and iterative. The parameters are

(i) the window width $W$ – the checking radius around the water pixel in question, defined in number of pixels (on Fig. 7 example $W = 1$); and

(ii) the number of iterations $L$ – how many times the algorithm must be applied over each water body (on Fig. 7 example $L = 2$).

Step 0 of the new algorithm starts working from the results of the basic flood-filling algorithm. In this case the basic flood-

filling algorithm should be applied so that it creates an individual water body mask, to avoid any mismatch between closely situated water bodies. Then the new algorithm may be applied to each water body successively. Step 0 is shown in Fig. 7 left plot. At Step 0, each water pixel is marked with "**x**" if all pixels within the moving window of the $W$ width are water, or "•" if at least one pixel in this window is non-water. Next starts the iteration phase, that will be repeated $L$ times. At the beginning of each iteration pixels with "•" are checked again with the moving window of the $W$ width – if around the pixel in question

there is at least one "**x**" pixel, it is marked as "••", see Fig. 7 second from the left plot. At the end of each iteration all "••" pixels are changed into "**x**", and the next iteration starts if required, see Fig. 7 third from the left plot. At the end of the iteration phase the considered water body will be divided into several ones, see Fig. 7 right plot – "**x**" pixels will mark the main part of the water body, and "•" pixels will mark the narrow rivers or bays. We applied this algorithm to separate automatically large rivers from the ocean – to stop deep penetration of the salt ocean into the land. The $W$ and $L$ parameters are regionally and grid

dependent. If they are unsuccessfully defined or coastal line is too complicated, the negative side effect of the algorithm will appear – erroneous separation of fjords and bays from the ocean (e.g. in Norway, North Canada, Greece and on the West coast of USA). To stay on the safe side all the separated water bodies with the area less than 500 km$^2$ were converted back to ocean. To minimize the tuning process, the new algorithm was applied only for the specific geographical locations, where big river estuaries and lagoon type freshwater lakes are situated, see Table 1. For the upgrade $L = 2$ and $W = 3$ were used. Figure 6 lower



row plots show results of basic flood-filling & newly developed pixel-by-pixel water separation algorithms use. The left plot in this row shows the region of Finland and North-Western part of Russia which looks the same as with use of the basic flood-filling algorithm only, because this region has no big river estuaries. The middle plot in the lower row shows the region of the St Lawrence River with neat separation of the freshwater river and saline ocean next to the Orleans Island in Quebec (Île d'Orléan). The right plot in the lower row shows the region of the Amazon River with the realistic separation of the ocean and river estuary.

Final step in the *LWM* water separation is the visual check of the significant freshwater coastal lagoons and lakes over the globe, in case if some separating islands or spits are missing on the initial ecosystem map. Also, some water bodies such as the Azov Sea and the Caspian Sea are better represented as inland water than ocean due to current features of IFS. This leads to a list of exceptional water bodies (see Table 2), that were manually separated from the ocean (the Caspian Sea is marked as a lake automatically), and creation of an updated *LKM*.

The upgrade of the *D_water* field concluded in combination of all most up-to-date reliable high-resolution global datasets, which are GLDBv3, ETOPO1 and GEBCO. Information from GLDBv3 is used for the mean depth of the inland water bodies, bathymetry of 36 large lakes and majority of Finnish lakes, ETOPO1 is used for the Great Lakes, and GEBCO is used for the Azov Sea, the Caspian Sea and the ocean bathymetry. The "default" 25 m depth was substituted with depth estimates based on geological approach (Choulga et al., 2014), which was implemented all around the globe. In rare cases where geological approach had no value, the "default" depth of 10 m was used. Figure 8 shows *D_water* field at 9 km horizontal resolution (Tco1279): the upper plot – the operational version, the lower plot – the new version. On average, all depths became shallower as the "default" depth of 25 m in the operational version was substituted with more realistic values.

The depth aggregation algorithm was also upgraded (from operational simple averaging). The lake depth is not a continuous field, like the air pressure or temperature, and averaging is not the most accurate way of treating it. The new lake depth aggregation is based on the mode (most common) value and the water type (ocean or inland water). Also, now the depth data source is considered: if there are in-situ measurements, indirect estimates or "default" value. For the depth aggregation only *LWM* water pixels are used; ocean and inland water pixels are aggregated separately. In the coastal regions, where both water types are present, *D_water* is averaged proportionally to the number of each water type pixels. Ocean pixels are aggregated by averaging as the ocean bathymetry can be considered as a continuous field (values change smoothly from point to point). For aggregation of the inland waterbody depths, the mode is used. The mode is calculated for each type of the depth data separately and the non-zero value with the highest priority is used as an aggregated grid-box depth; the highest priority is given to the value calculated only from the in-situ measurement, the second – to the value calculated only from the depth indirect estimates, the lowest – to the "default" 10 m depth. This helps to preserve the measured values at rather high resolutions where the lake effect is the most pronounced.





## 4 Verification and discussion

Upgraded lake related fields must be tested prior operational implementation, as inland water bodies can have significant impact on local climate and weather in terms of 2-meter temperature: over 1 K (Balsamo et al., 2012) and up to 10 K (Eerola et at., 2014) respectively. FLake prognostic variables are: the mixed-layer temperature $T_{ML}$, the mixed-layer depth, the bottom

temperature, the mean temperature of the total water column, the shape factor, the temperature at the ice upper surface, and the ice thickness (IFS Documentation, 2017). Verification is performed in terms of $T_{ML}$ and the ice formation/disappearance dates. Modelling results are verified against in-situ measurements of lake water surface temperature and ice formation/disappearance dates recorded by Data and Information Centre of the Finnish Environment Institute (SYKE).

### 4.1 Model experiment setup and verification methods

Numerical experiments with IFS model using operational and upgraded *LKM* and *D_water* run for 5 years from 2010.01.01 to 2014.12.31, with 3 months of model spin up from 2009.10.01 to 2009.12.31. Experiments started in the middle of autumn 2009, when all Finish lakes are mixed till the bottom, to shorten the model spin up time and to get reliable results straight after ice melting in spring of 2010. Experiments run with IFS CY43R3 model on the triangular cubic octahedral grid with the high horizontal resolution ~ 9 km (i.e. Tco1279), in the surface off-line mode (i.e. no feedback of the surface to the atmosphere).

For the forcing, the lowest model level variables were taken from the newly available ERA5 reanalysis (C3S, 2017). In ERA5, the lake parameterization is included in the model. The experiments *GTZP_OPR* (red in all figures) and *GTZL_NEW* (blue in all figures) used operational and upgraded *Fr_lake* and *D_water* values respectively.

For verification, we used the standard scores: mean error or bias (difference between observed and simulated values), MAE, and standard deviation error (STD). The statistical significance of the difference in model errors between two experiments are

checked with a non-parametric Kruskal-Wallis test (Glantz, 2012) as prior it was noted that errors have non-Gaussian distribution. For the Kruskal-Wallis test, data from all comparing groups are combined, sorted ascending and ranked, equal values are assigned with their mean rank. The Kruskal-Wallis test statistic $H$ is:

$$H = \frac{12}{N(N+1)} \sum_{k=1}^{K} [n_k (\overline{R_k} - \bar{R})^2], \tag{4}$$

where $K$ – number of groups, $n_k$ – sample volume for the group $k$, $N$ – the total volume of all groups combined $N = \sum_{k=1}^{K} n_k$,

$\overline{R_k}$ – the average rank of the group $k$, $\bar{R}$ – the average rank of combined groups $\bar{R} = \frac{N+1}{2}$. To estimate the statistical significance, $H$ is compared with a critical value $\chi^2$ for $(K-1)$ groups with the significance level $\alpha$ (if not stated differently $\alpha = 0.05$). If $H > \chi^2$, then differences between groups are statistically significant.

In-situ SYKE data. SYKE is responsible for producing, storing and distributing Finland's national environmental information and spatial data (SYKE, 2017). SYKE operates more than 30 regular lake and river water temperature measurement sites over

Finland. In-situ lake water surface temperature measurements and on-shore observations of the lake visible area freeze-up/break-up dates collected by SYKE are used for the model verification. The water temperature is measured every morning during the ice-free season at 8.00 am local time, close to shore, at 20 cm below the water surface (Rontu et al., 2012, Kheyrollah





Pour et al., 2017). Temperature measurements and ice formation/disappearance dates from 27 lakes for 2010-2014 are used for verification. Locations of the measurement points are shown in Fig. 9.

The main morphological properties of lakes are given in Table 3 and Fig. 10. This table contains also *D_water* values from the model grid. Differences between in-situ depth measurements and *D_water* values from the model are due to horizontal resolution: the in-situ depth values are from point measurements and the model depth values are from aggregated 9 by 9 km grid-boxes. During the *D_water* upgrade it was noted that lake Saimaa has incorrect mean depth (18.0 m instead of 10.8 m), correction is planned during the next upgrade.

Comparison between the operational and upgraded fields, considering the error as a difference between in-situ and modelled values, shows that for 27 selected lake sites even with 9 km resolution the upgraded *D_water* values have 25.4 times lower bias (-0.2 m instead of -4.8 m), 3.4 times lower MAE (2.4 m instead of 8.2 m) and 2.7 times lower STD (3.6 m instead of 9.7 m). Changes are statistically significant.

## 4.2 Model verification results

Measured and modelled lake surface temperatures were compared for the full experiment period 2010-2014. The model values were sampled for the ice-free season at 8.00 am local time to correspond the measured values. Figure 11 shows the bias, MAE, STD and total amount of data used per each site. In general, errors became smaller (modelled values are closer to the measured ones) as the lake depth values became more realistic. Averaging over all 27 lakes, the comparison between two experiments shows that for $GTZL_{NEW}$ bias is lower for 12.5 %, MAE – for 13.4 %, and STD – for 20.3 %. For some lakes water temperature modelling errors remained the same as their depth values are the same or changed insignificantly in two experiments. These lakes are: Paijanne, Pyhajarvi, Paajarvi2, Kuivajarvi, Pesiojarvi, Rehja-Nuas, Kilpisjarvi and Inarijarvi. The only statistically significant deterioration in the temperature scores was for lake Lappajarvi, which depth is overestimated 2.5 times in the upgraded *D_water* (18.0 m instead of 6.9 m) due to the depth mapping algorithm and/or horizontal resolution of the depth field.

Model errors may be different during different seasons depending on the model physics. It was shown that FLake has the best performance in boreal zone during autumn, when lakes are mixed (Choulga and Kourzeneva, 2014), provided that the lake depth is correct. Thus, it is interesting to dig into details and to verify the model results for different seasons, depending on lake mixing regime. Typically, lakes in the boreal zone are dimictic (Lewis, 1983) and have 4 main seasons in relation to mixing and ice cover:

(i) spring mixing, when lakes are mixed till the bottom and the mixed layer depth equals to the lake depth,

(ii) summer, which is the stratified period,

(iii) autumn mixing period, which is usually longer than spring, and

(iii) winter, when lakes are covered with ice.

However, this classical pattern is approximate, it may be distorted, depending on the lake depth and the atmospheric forcing. For example, a stratified summer period may be interrupted by a short mixing period. Also, in early spring the inverse





temperature stratification may appear. Patterns of mixing and ice periods may be defined from the modelling results. Figure 12 shows ice covered (blue), mixed (red) and stratified (green) periods, defined for different lakes for the model experiments $GTZP_{OPR}$ and $GTZL_{NEW}$. Most of the selected lakes show rather complex behaviour with distorted classical pattern. For example, lakes Paajarvi2 and Kuivajarvi may have the multiple ice and mixing periods during the year. Some lakes change

patterns from one experiment to another, because of noticeable depth changes (e.g. Haukivesi, Saimaa). To ease the verification process, these patterns were smoothed to better correspond with the dimictic lake classical pattern. For each lake in both experiments, each year was separated into 4 main uninterrupted lake seasons, according to the modelling results. Figure 13 shows the results:

(i) winter period (blue), which contains the merged ice periods when ice-free time between them is 30 days or less,

(ii) spring and autumn mixing periods (red and yellow respectively), which contain the merged mixed periods (when the mixed layer depth is approximately equal to the lake depth, with the maximum difference of 10 cm allowed) when the stratified regime between them is 20 days or less,

(iii) the stratified summer period (green), which is defined as a residual between spring and autumn periods.

Thus, the spring and autumn mixing periods appeared to be separated by the summer stratified period (e.g. lake Inarijarvi).

With this approximation, some lakes became monomictic (Lewis, 1983), containing no stratified period (e.g. lakes Pyhajarvi and Tuusulanja). For the verification purposes, for these lakes the mixing period was equally divided between spring and autumn seasons.

Distribution of model errors in terms of $T_{ML}$ depending on a mixing season is shown in Fig. 14-17. The important note is that bias in both experiments in all seasons was predominantly cold (positive) and large. It was so large that STD was smaller than

20 bias. In another FLake model error studies bias was dependent on the season. For example, in (Kourzeneva, 2014), where forcing was from the High Resolution Limited Area Model (HIRLAM) (Unden et. al, 2002), in summer for the same Finish lakes there was a strong warm bias, while in spring bias was cold. Errors in $T_{ML}$ simulations depend on FLake itself, on the errors in $D\_water$, which is the main lake model parameter and on the errors in forcing. Since results of current experiments differ from the other studies, it should be suggested that in present research errors came from the forcing – ERA5 is supposedly

too cold for this region. This problem was previously mentioned in (Haiden et al., 2018). Thus, for $D\_water$ parameter, the situation of compensating errors may appear, depending on a season. Too shallow (underestimated) lake depth can lead to a smaller cold bias during spring mixing and a stronger cold bias during autumn, while the overestimated $D\_water$ parameter can lead to a stronger cold bias in spring and a smaller bias in autumn. In other words, for better spring results it is "advantageously" to underestimate $D\_water$, but for better autumn results it is "advantageously" to overestimate it. In the

stratified summer period, this kind of compensation does not take place, because the mixed layer depth during stable stratification does not depend on the lake depth. However, in summer $T_{ML}$ diurnal cycle depends on $D\_water$: the deeper the lake, the smaller the $T_{ML}$ diurnal cycle amplitude. This may be reflected in STD scores because they relate to the diurnal cycle amplitude in present experiments. These suggestions are in accordance with the obtained results. For all lakes, where upgraded $D\_water$ was smaller than operational one, $GTZL_{NEW}$ bias was smaller in spring and larger in autumn comparing with $GTZP_{OPR}$



(e.g. lakes Konnevesi and Vaskivesi). And vice versa, for all lakes, where upgraded $D\_water$ was larger than operational one, $GTZL_{NEW}$ bias was larger in spring and smaller in autumn comparing with $GTZP_{OPR}$ (e.g. lakes Haukivesi and Oulujarvi). This was independent from whether new $D\_water$ is closer to the reality or not. For example, for lakes Lappajarvi and Saimaa, where upgraded $D\_water$ became larger and even further from the reality than operational, $GTZL_{NEW}$ autumn bias improved,

due to compensating errors (good result for the wrong reason). The only exception was lake Niilakka, which autumn bias was negative (warm). For the combined spring-autumn mixing period, bias scores were generally better, or the effect was neutral. For the summer stratified period, the impact of $D\_water$ on the bias scores was neutral or slightly positive. The STD scores were the best for the autumn mixing period, when the lake surface temperature diurnal cycle is absent. For lakes Saimaa and Lappajarvi, the summer period STD scores were worse in $GTZL_{NEW}$ comparing with $GTZP_{OPR}$, however $D\_water$ was worse

as well. For the lakes with better $D\_water$ values in $GTZL_{NEW}$, STD scores improved or remained unchanged for all seasons. Exception was lake Oulujarvi, its STD scores deteriorated, mainly in autumn.

Winter season verification was based on ice formation/disappearance dates comparison. Table 4 shows the ice formation/disappearance dates from SYKE in-situ archive and based on experiment results with operational ($GTZP_{OPR}$) and upgraded ($GTZL_{NEW}$) $D\_water$ for 27 lake sites. In general, present experiments showed too late ice melt in spring and too

early ice formation in autumn, this is in accordance with suggestion of a cold bias in forcing. Thus, compensation may happen also for the errors in freeze-up dates: for compensating the cold forcing, it is "advantageously" to overestimate the lake depth. Melting dates are mainly dependent on the atmospheric forcing rather than $D\_water$, but for the freeze-up dates $D\_water$ plays an important role. For the melting dates it was almost no difference between two experiments, but in freeze-up dates the difference was substantial. Errors were large – ice melt date maximum error was 26 days (lake Niilakka in 2011) and ice

freeze-up date maximum error was 61 days (lake Oulujari in 2017, $GTZP_{OPR}$). The ice off date errors were not dependent on $D\_water$, the largest errors corresponded to large-area lakes (e.g. lakes Haukivesi and Kallavesi, see Table 4). It can be explained by the fact that the ice formation/disappearance in-situ measurements represent the freeze-up and break-up dates in the visible area around the observer's location (usually on the shore), and due to physiographic features (e.g. complicated rugged coast) and/or meteorological conditions (e.g. low clouds, rain) can be not fully representative for the whole 9 by 9 km

grid-box. Ice measurement locations differ from temperature measurement locations, and distance between these two can vary from 0.7 to 49.0 km, see Table 5. SYKE provides also the break up dates in far central parts of the lake and permanently freeze up dates of the visible area around the observer's location, but the amount of data is very limited and can't be used for verification. However, it gives a hint that in the central part of a lake comparing with the shore, ice breaks later up to a week, and close to the coast the permanent ice can appear straight away or up to even a month after the first freeze-up date. The

rough estimate of the error due to the model and forcing comes from the break-up date analysis for Lake Kevojarvi. This lake has small representativeness error, because its surface area is only 1 km². However, the error in the break-up date for this lake was large – 14 days in both $GTZP_{OPR}$ and $GTZL_{NEW}$ experiments. Thus, in this verification no difference between experiments $GTZP_{OPR}$ and $GTZL_{NEW}$ were assumed, if it was less than 14 days. In Table 4, improvements in the freeze-up date in $GTZL_{NEW}$ comparing with $GTZP_{OPR}$ are marked with bold and underline, and degradation – with italics and underline, but only for the





cases, when difference was larger than 14 days. Otherwise no impact of *D_water* is considered. From Table 5, freeze-up dates improved for the lakes with increased *D_water*, these lakes became deeper and start to freeze later (e.g. lakes Oulujarvi and Unari). This is independent from whether new *D_water* is closer to the reality or not (e.g. for lakes Saimaa and Lappajarvi, the freeze-up dates improved for wrong reasons). If during the upgrade *D_water* decreased, errors became larger (e.g. lakes Konnevesi and Vaskivesi). This agrees with the autumn $T_{ML}$ bias scores: if they improve, the freeze-up dates improve as well.

### 4.3 Discussion

Upgraded lake related fields were tested for 5 successive years to capture short climate deviations (one particular year can be slightly warmer or colder than the average one), yet not to deal with major water distribution and/or inland water body depth changes that can occur in 10-year period and would have to be taken into account comparing against in-situ measurements. Current verification included only 27 lake sites over Finland which are freely available online, it would be useful to compare model results with measurements from the other countries and climate zones as IFS is a global forecasting system. Experiments run with model cycle CY43R3. New cloud physics in cycle recent upgrade lead to improvements in calculating 2-meter temperature and humidity, and precipitations (especially near coasts) which can lead to better agreement of the modelled and in-situ lake surface temperature and ice formation/disappearance dates respectively. Verification of operational and upgraded *D_water* for 27 Finish lakes resulted in significant reduction of errors, though it is still possible to upgrade *D_water* with new measurements and test new aggregating techniques in order to better represent initial high-resolution lake depth field. Verification in terms of modelled and in-situ lake surface temperature for the whole 5-year period showed general error reduction for 12-14 %. Seasonal verification also showed overall error reduction although amount of data during 5-year period was not sufficient to have always statistically significant results. Seasonal verification also revealed the cold bias in the forcing and situation, when changes in the *D_water* parameter compensate this bias. For more detailed ice formation/disappearance date verification and explanation of the results, first and permanent ice formation/disappearance dates in a far central part of the lake (compatible with IFS model high resolution 9 km grid), are needed.

### 5 Conclusion

Earth System Models used for weather and climate monitoring and forecasting applications, including the IFS, need lower boundary conditions (skin temperature, surface fluxes of heat, moisture and momentum) to calculate the evolution of dynamic processes in the atmosphere and to produce a usable weather forecast. To compute them sufficiently accurately, an up-to-date ecosystem map is needed. Nowadays human activities influence Earth's surface and adapt it to societal needs on relatively short timescales, for example to construct new artificial lakes to supply people and/or crops in arid places with water, or to create new islands to build homes. Inland water bodies can influence local climate by over 1 K (Balsamo et al., 2012), and the influence on local weather can be even more pronounced: correct lake surface state (ice/no ice) in winter conditions can lead to up to 10 K difference in 2-meter temperature (Eerola et at., 2014). Major changes in water bodies can occur in just a few



years, which means that ecosystem-based maps used for numerical weather prediction need to be updated regularly. The most frequent updates of ecosystem maps come from satellite products, which are becoming available at increasingly high resolution. The main obstacle to using these maps in the model without any modification is that they do not distinguish between ocean and inland water. An automatic algorithm to separate ocean and inland water has been presented in this article. This

algorithm can also be used to distinguish between rivers and lakes, but it will require more testing and tuning of parameters before it can be applied globally. For the IFS, the most reliable data sources are used to ensure the best possible representation of the global inland water distribution. The continuous water depth field is regularly updated with new ocean and lake bathymetries, new versions of the lake database, and indirect depth estimates based on the geological origin of lakes. Verification of the depth field for 27 Finish lake sites showed significant lake depth error reductions in the GLDBv3 dataset

compared to GLDBv1. Verification in terms of the lake water surface temperature showed an overall error reduction of between 12 and 14 %. Seasonal lake water surface temperature verification, according to lake mixing periods (spring mixing, summer stratification and autumn mixing) showed an overall error reduction, although forcing in the numerical experiments was too cold, and it may be that this error was compensated for by lake depth parameter errors. Winter season verification based on an ice formation/disappearance date comparison was also influenced by the problem of too cold forcing and compensating errors.

A more detailed ice formation/disappearance date verification and further experiments are clearly needed. The first and permanent ice formation/disappearance dates in far central part of the lake (compatible with IFS model high resolution 9 km grid) would be very helpful for verification. Lake depth and lake cover variability over time are recognised as key aspects for future developments. The present study aims to document the methodology and to provide experimental evidence of its benefits, and it will be used to characterise temporal variations (e.g. in annual or monthly updates).

**Data availability**

SYKE datasets are freely available online at http://rajapinnat.ymparisto.fi/api/Hydrologiarajapinta/1.0/. ERA5 reanalysis is freely available online at http://www.ecmwf.int. Source code of lake model FLake is freely available online at http://www.lakemodel.com. Raw output of the IFS model at 9 km resolution for 27 verification sites is available from the corresponding author by request.

**Author contribution**

All authors were participating in lake field update (methodology, data generation), IFS model experiment set up, analysis of the in-situ and model result comparison. Margarita Choulga wrote the manuscript with contributions from all other authors.



**Competing interests**

The authors declare that they have no conflict of interest.

**Acknowledgements**

The authors thank Matti Horttanainen (FMI) for providing in-situ $T_{ML}$ and ice formation/disappearance dates from the SYKE

archive; Laura Rontu (FMI) for useful discussions and help with data handling; Emily Gleeson (Met Eireann), Peter Janssen (ECMWF) and Joe McNorton (ECMWF) for editorial help and assistance. Margarita Choulga was funded by the Earth2Observe project which received funding from the European Union's Seventh Programme for research, technological development and demonstration under grant agreement No 603608.

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

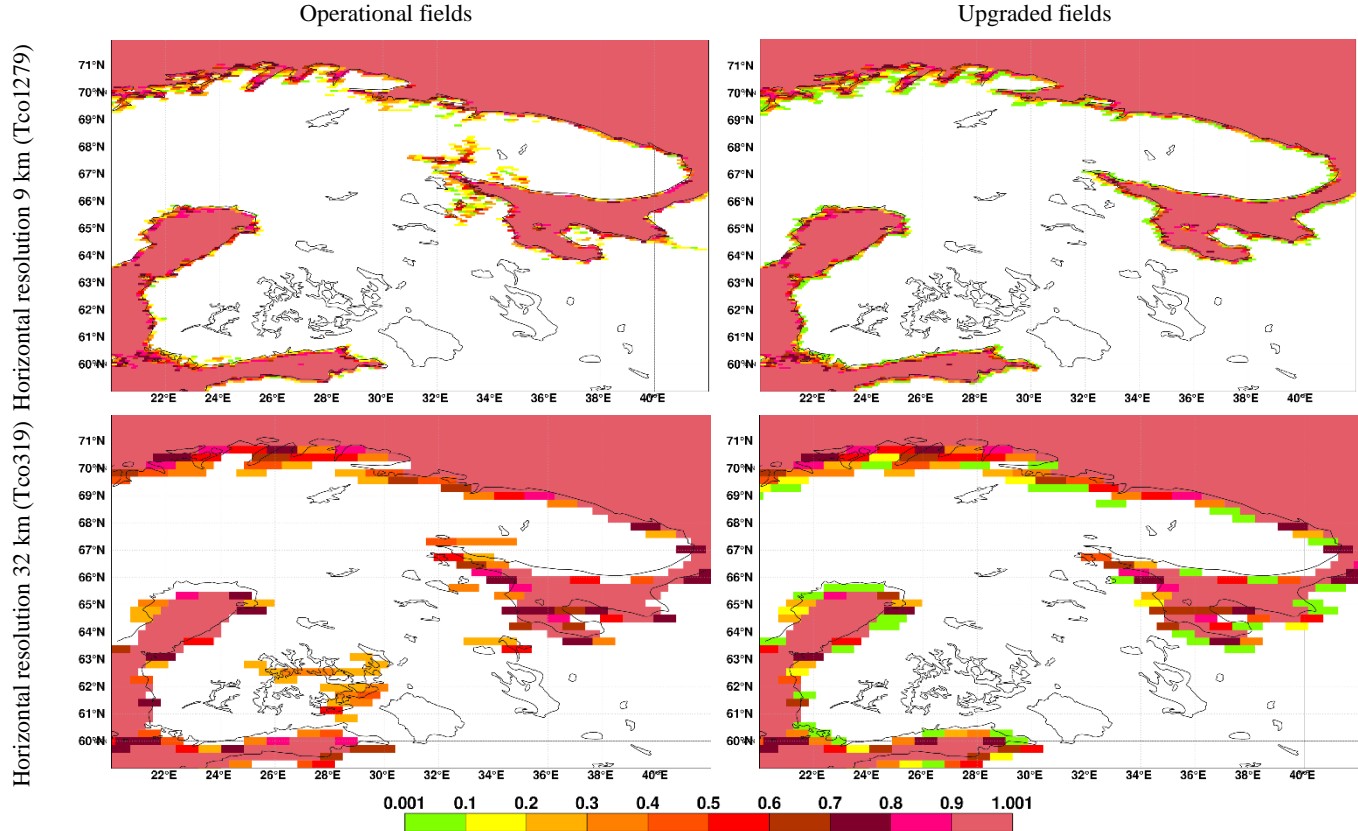

**Figure 1: Combination of operational and upgraded *Fr_land* and *Fr_lake* fields showing remaining ocean water over Finland and North-Western Russia (59-72 °N, 20-42 °E) at different horizontal resolutions; colours indicate ocean fraction in each grid-box: white - no ocean, pink - fully covered with ocean**



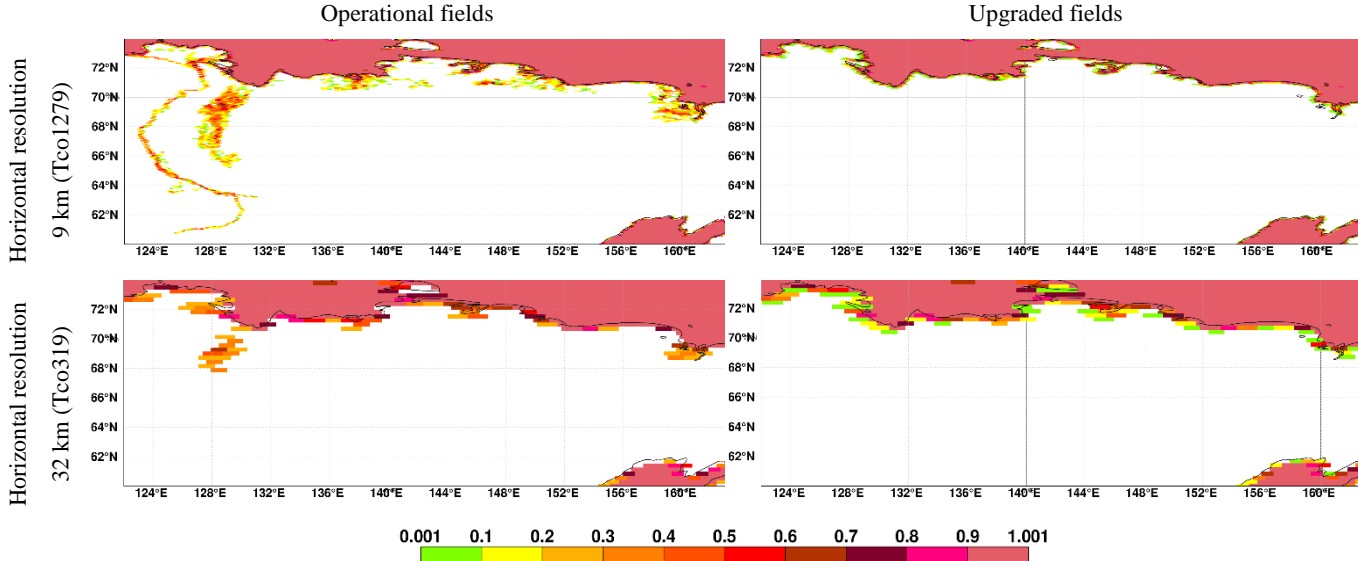

Figure 2: Same as Fig. 1, but over North-Eastern Russia (60-74 °N, 122-163 °E)

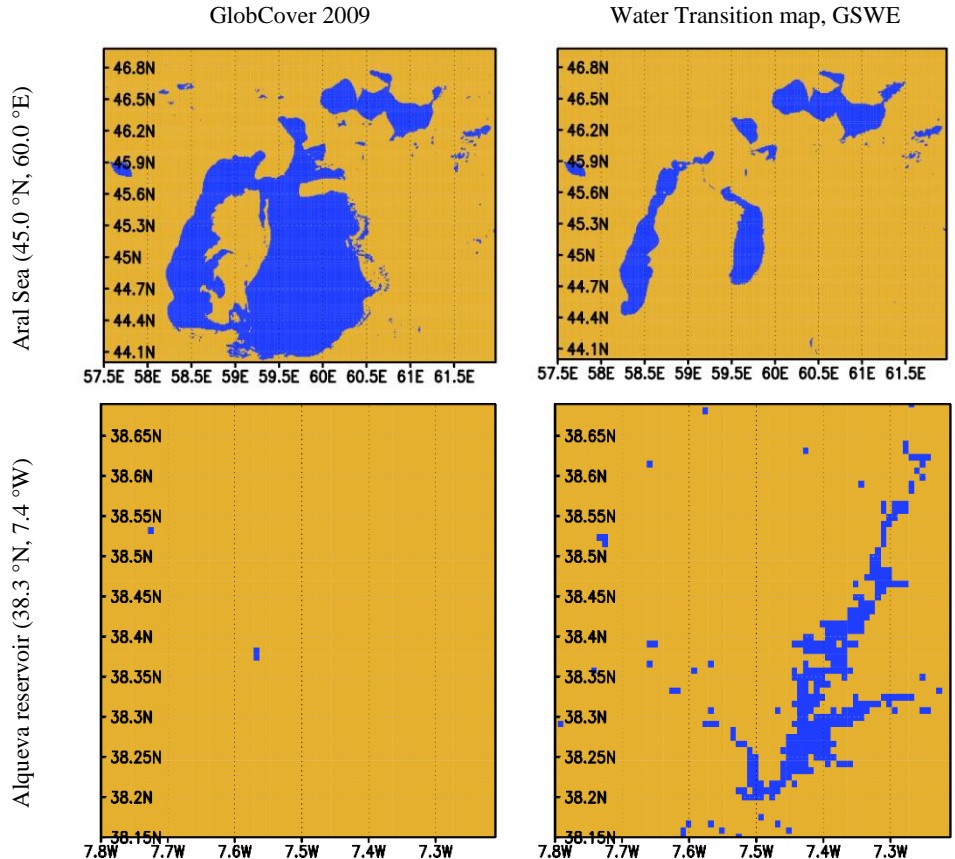

**Figure 3: Water distribution from GlobCover 2009 and GSWE Water Transitions map (only (1) Permanent, (2) New Permanent and (7) Seasonal to Permanent water classes are used); yellow colour indicates land, dark blue – water**



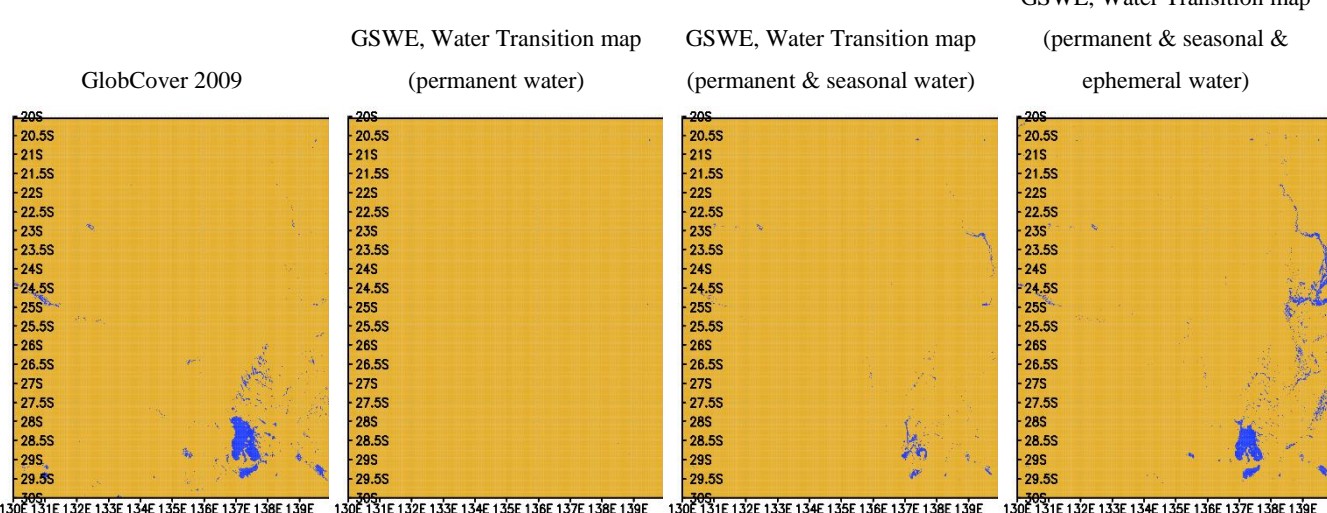

**Figure 4: Water distribution for Australian (20-30 °S, 130-140 °E) region using GlobCover 2009 and GSWE Water Transition map with different water classes combinations; permanent water stands for combination of (1) Permanent, (2) New Permanent and (7) Seasonal to Permanent water classes; seasonal water – (4) Seasonal, (5) New Seasonal and (8) Permanent to Seasonal; ephemeral water – (9) Ephemeral Permanent and (10) Ephemeral Seasonal; yellow colour indicates land, dark blue – water**

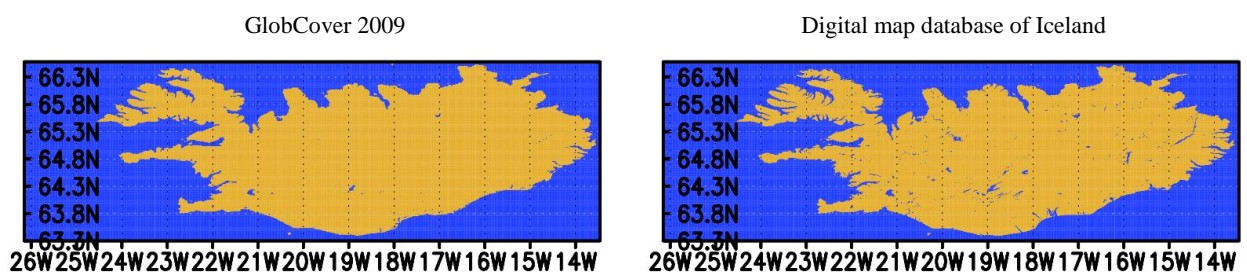

**Figure 5: Water distribution for Iceland using GlobCover 2009 and Digital map database of Iceland; yellow colour indicates land, dark blue – water**



**Figure 6: Phases of *LWM* water separation for Finland and North-Western part of Russia (left column), St Lawrence River region (middle column), and Amazon River region (right column): no water separation (upper row), separation with flood-filling algorithm only ("basic" flooding, middle row) and separation with flood-filling and newly developed pixel-by-pixel water separation algorithms ("extra" flooding, lower row); yellow colour indicates land, dark blue – inland water (in middle and lower row) or total water (in upper row), light blue – ocean**



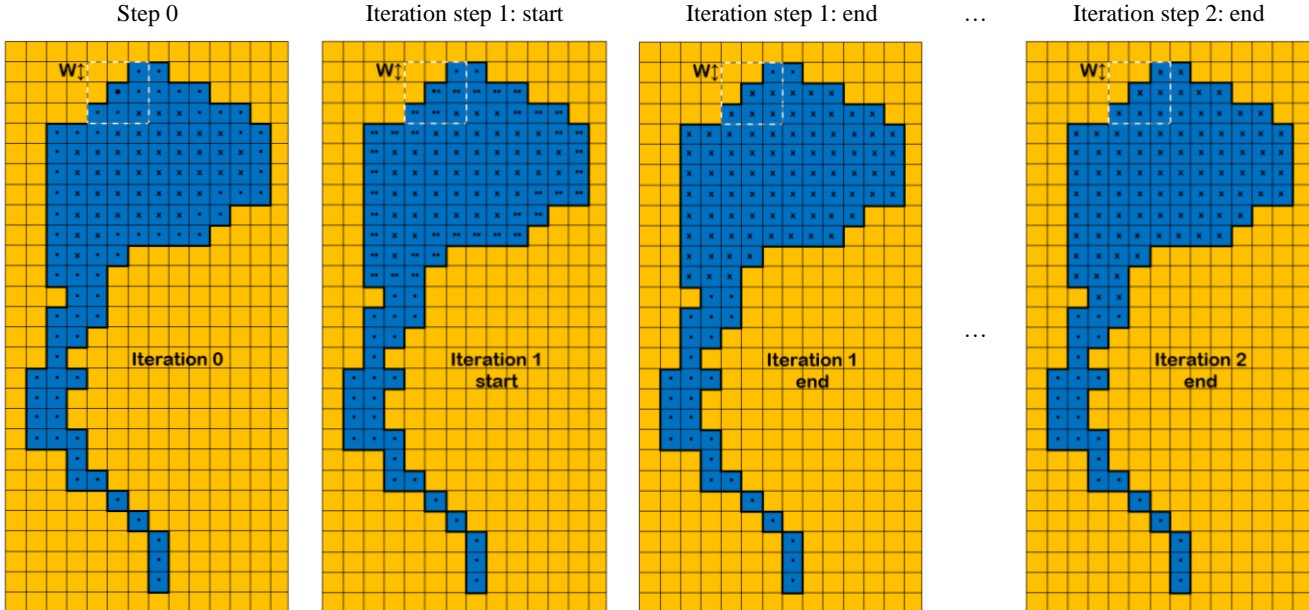

**Figure 7: Steps of pixel-by-pixel water separation algorithm; L – number of iterations (here L=2), W – window width (here W=1),**
**• – water grid-box has no water points in its checking window, x – water grid-box has only water points in its checking window,**
**•• – water grid-box has at least one x in its checking window; yellow colour indicates land, dark blue – water**

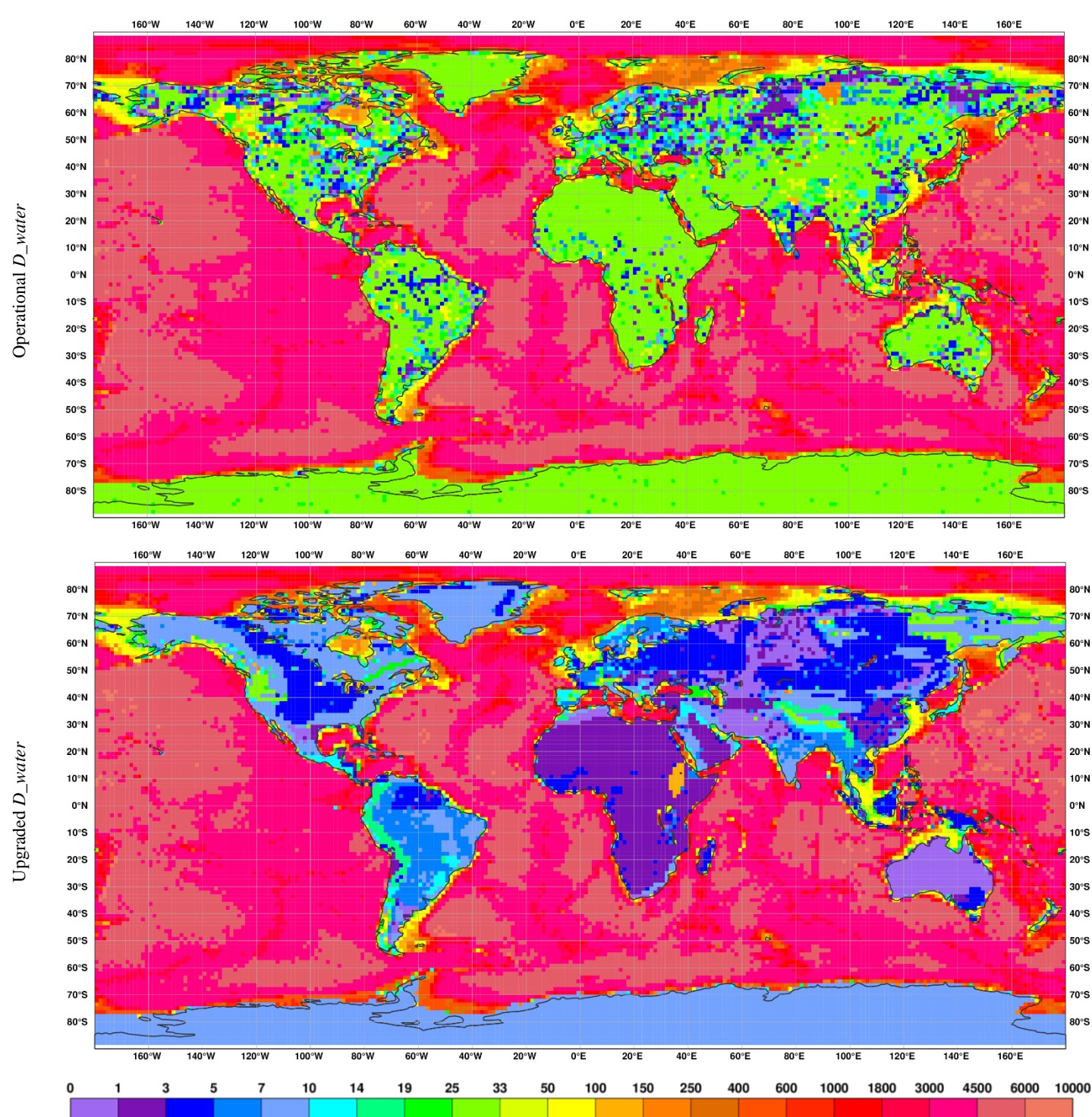

**Figure 8: Operational (upper panel) and new (lower panel) depth fields at 9 km horizontal resolution (Tco1279); depth values in meters**



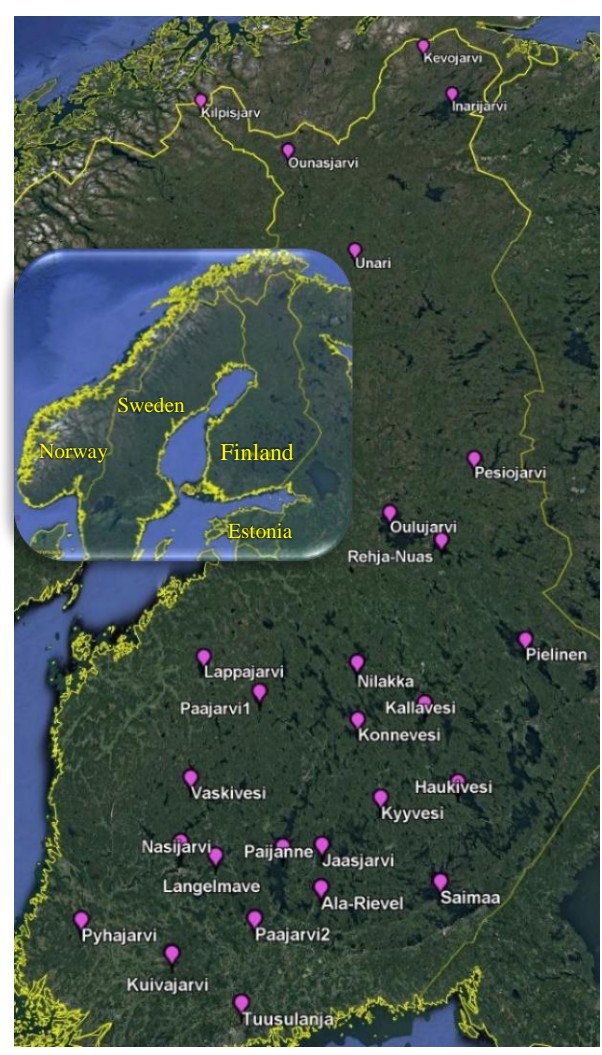

**Figure 9: Locations of 27 lake verification sites (Google Maps, maps.google.com, 2019)**





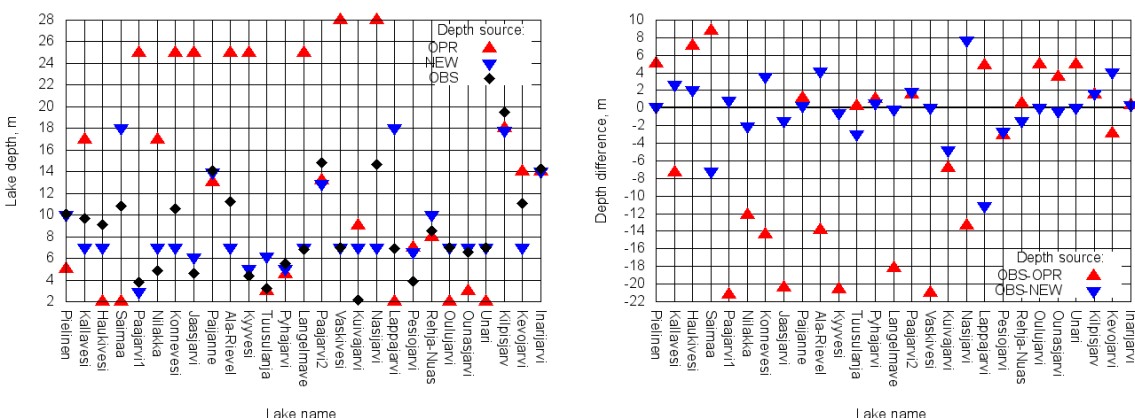

**Figure 10: Lake depths and their differences in meters for 27 verification sites; OBS – measured by SYKE, OPR – from ECMWF operational file and NEW – from upgraded file**

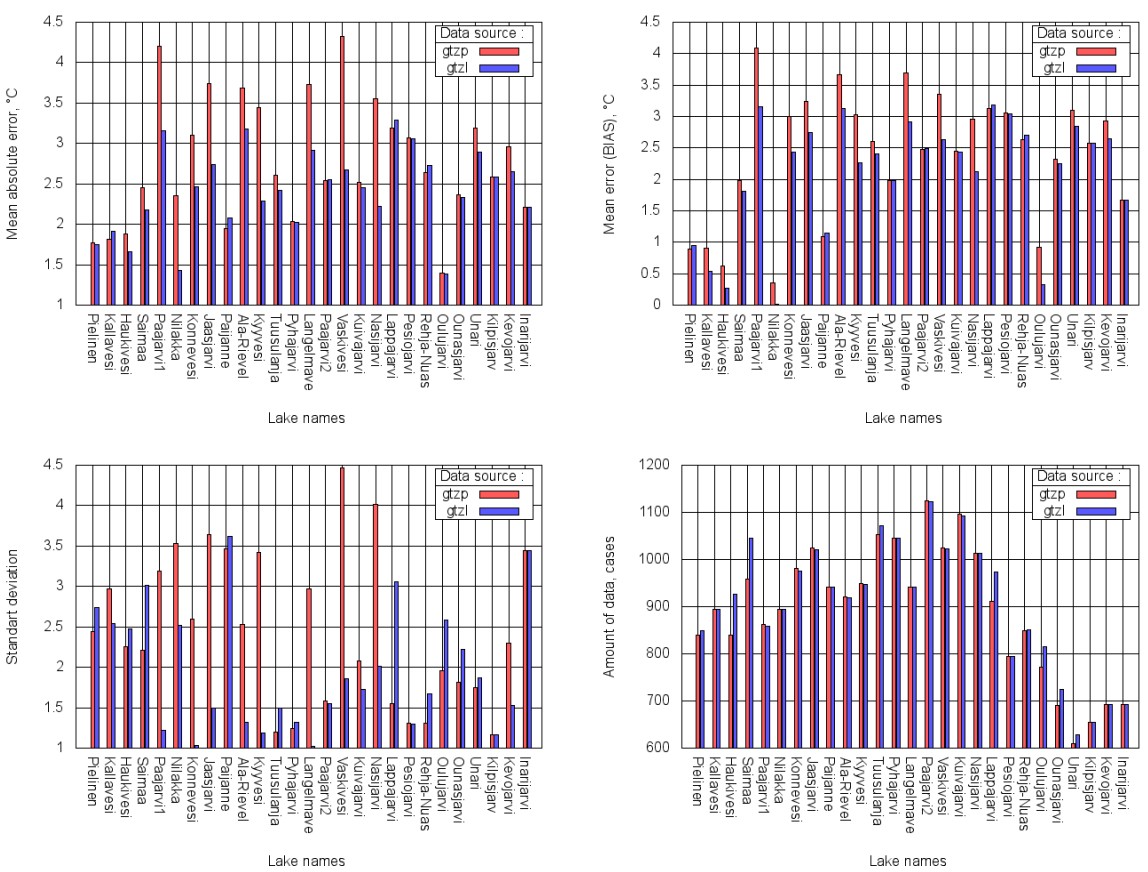



**Figure 11: MAE, bias, STD and amount of data calculated over the total period of 2010-2014 for 27 verification sites; *GTZP* (red) – experiment with operational *D_water*, *GTZL* (blue) – with upgraded *D_water***

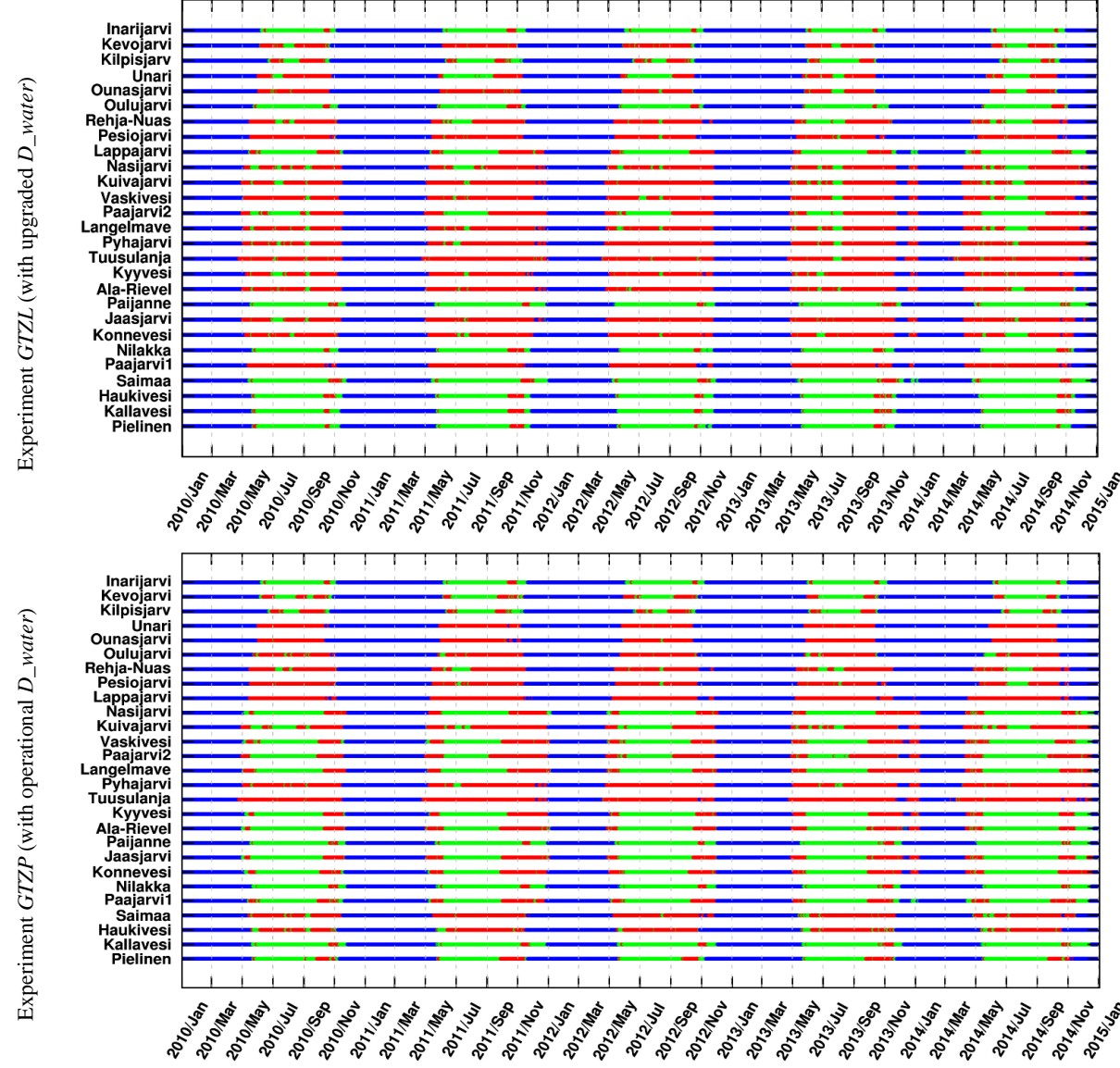

**Figure 12: Lake seasons for 2010-2014 for 27 verification sites based on operational (lower panel) and upgraded (upper panel)**
*D_water*; **blue – lake is ice covered, red – lake is mixed till the bottom, green – lake is stratified (ice-free and non-mixed, residual period)**

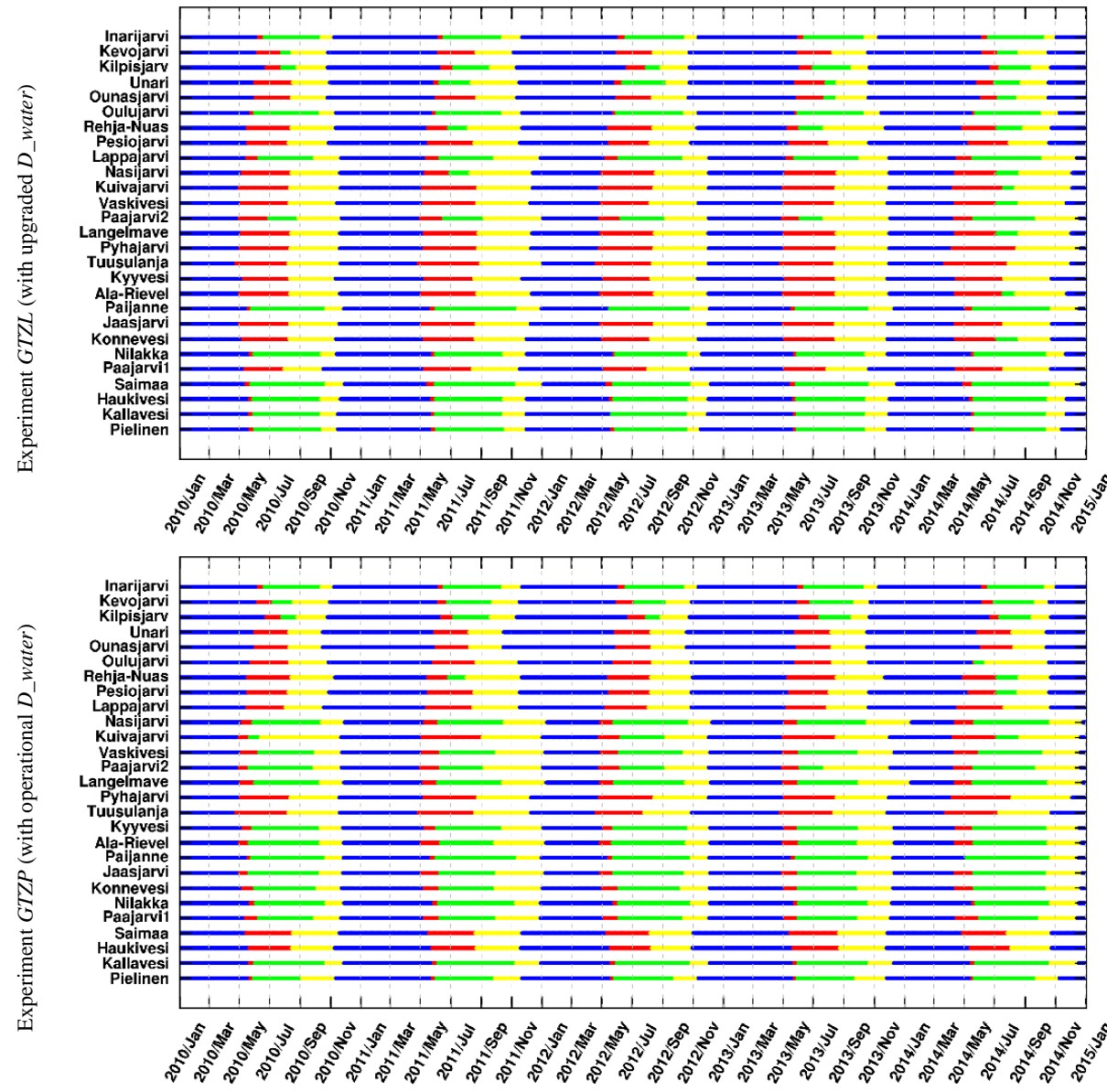

**Figure 13: Uninterrupted lake seasons for 2010-2014 for 27 verification sites based on operational (lower panel) and upgraded (upper panel) *D_water*; blue - winter, red - spring mixing, green - stratified summer, yellow - autumn mixing period**





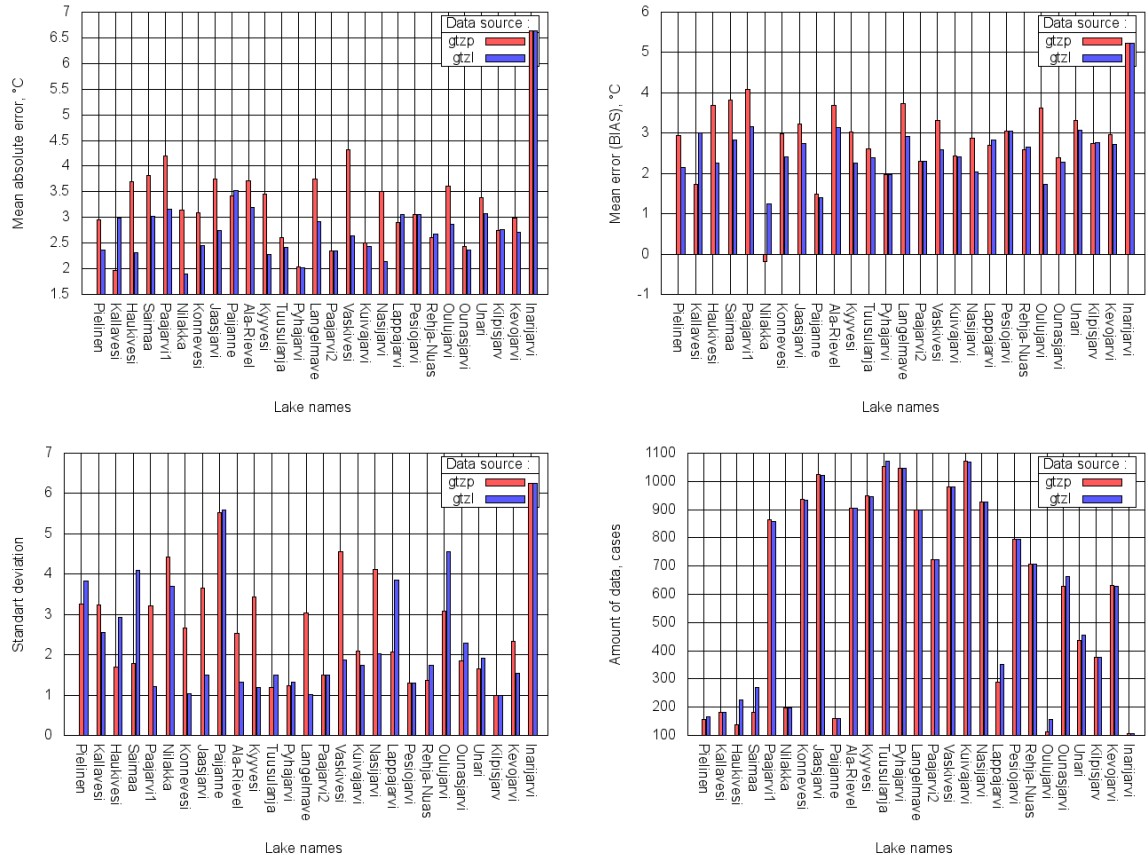

**Figure 14: MAE, BIAS, STD and amount of data calculated over all mixing periods 2010-2014 for 27 verification sites; *GTZP* (red) – experiment with operational *D_water*, *GTZL* (blue) – with upgraded *D_water***





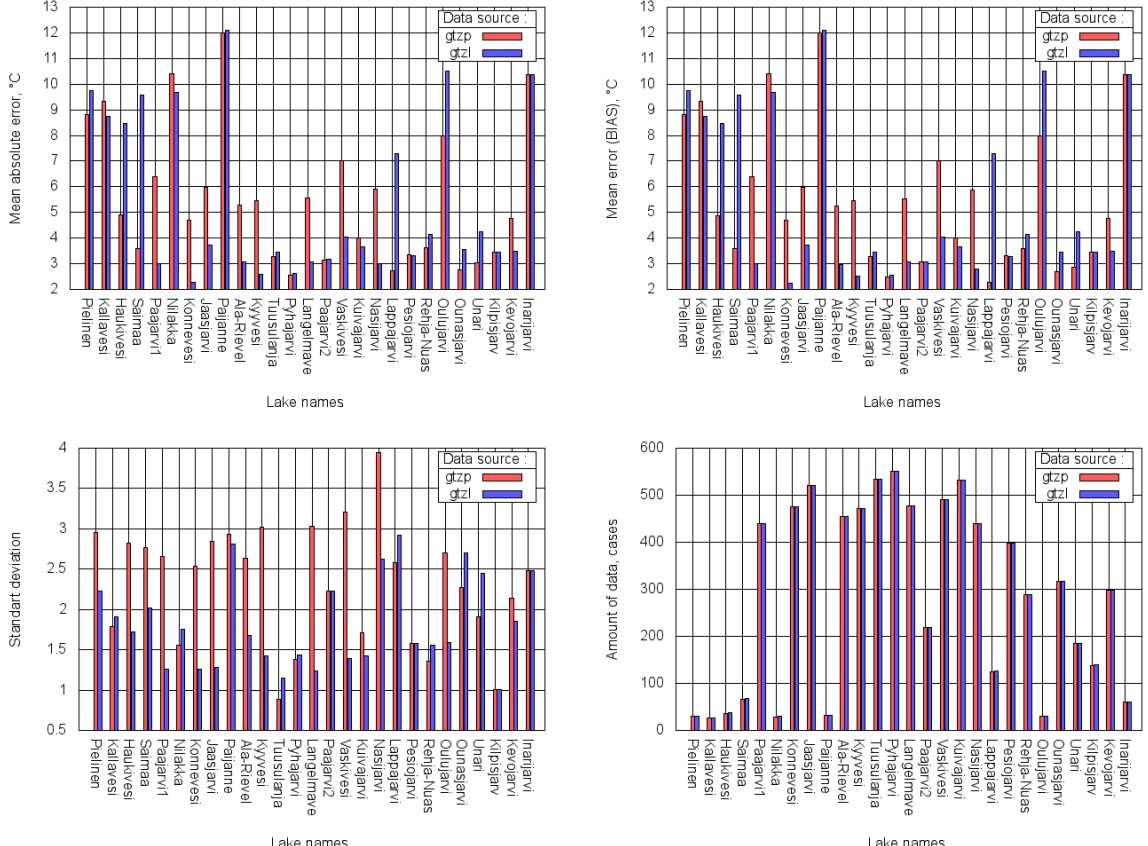

**Figure 15: Same as Fig. 14, but calculated over all spring mixing periods**



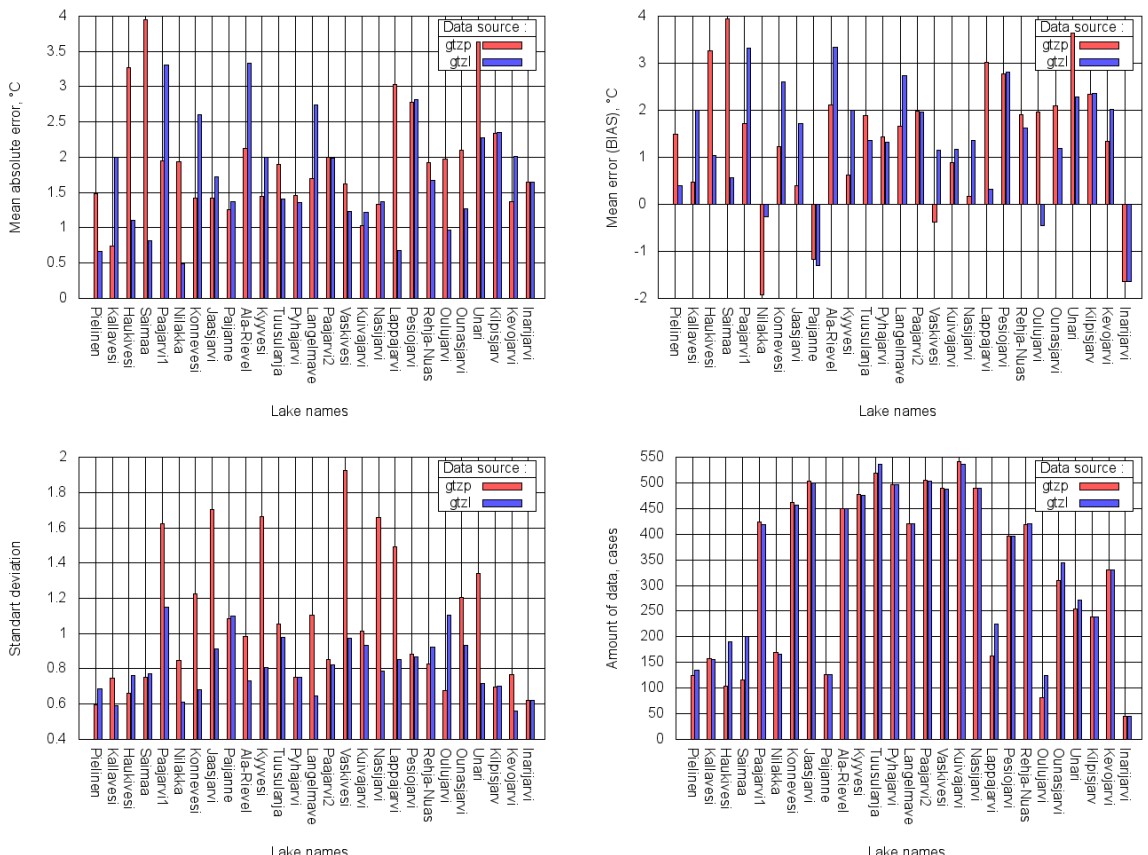

**Figure 16: Same as Fig. 14, but calculated over all autumn mixing periods**





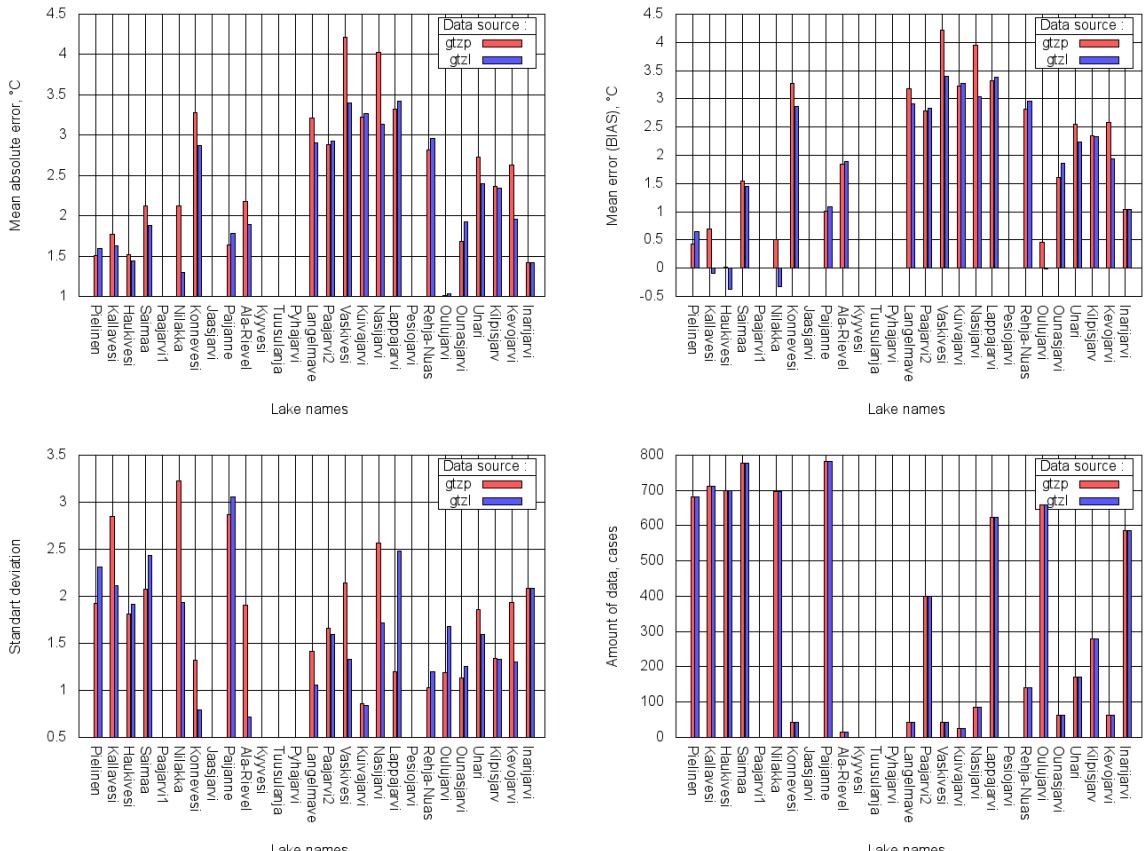

**Figure 17: Same as Fig. 14, but calculated over all stratified summer periods**





**Table 1: List of geographical locations for the water pixel-by-pixel separation algorithm application**

| Region № | North | South | West | East | Region № | North | South | West | East |
|---|---|---|---|---|---|---|---|---|---|
| 1 | 80°N | 70°N | 80°E | 90°E | 12 | 34°N | 30°N | 120°E | 130°E |
| 2 | 80°N | 70°N | 120°E | 130°E | 13 | 30°N | 20°N | 100°W | 90°W |
| 3 | 70°N | 60°N | 170°W | 160°W | 14 | 30°N | 0°N | 90°E | 100°E |
| 4 | 70°N | 60°N | 140°W | 130°W | 15 | 20°N | 10°N | 70°E | 80°E |
| 5 | 70°N | 60°N | 60°E | 70°E | 16 | 20°N | 10°N | 100°E | 110°E |
| 6 | 70°N | 60°N | 160°E | 170°E | 17 | 10°N | 10°S | 60°W | 50°W |
| 7 | 60°N | 50°N | 60°W | 50°W | 18 | 0°N | 10°S | 80°W | 70°W |
| 8 | 60°N | 50°N | 10°E | 20°E | 19 | 0°N | 10°S | 50°W | 40°W |
| 9 | 60°N | 50°N | 140°E | 150°E | 20 | 0°N | 10°S | 10°E | 20°E |
| 10 | 50°N | 40°N | 80°W | 60°W | 21 | 0°N | 10°S | 140°E | 150°E |
| 11 | 40°N | 30°N | 90°W | 80°W | 22 | 30°S | 40°S | 60°W | 50°W |

**Table 2: List of the exceptional water bodies for manual separation from the ocean**

| Latitude | Longitude | Water body name | Reason for separation |
|---|---|---|---|
| 46.06°N | 36.64°E | The Azov Sea | Currently better modelled as inland water than ocean |
| 68.66°N | 53.01°E | Lake Peschanka | Freshwater shallow lake with surface area 122 km$^2$ |
| 16.31°N | 94.90°W | Laguna Superior | Lagoon with surface area 380 km$^2$ |
| 10.17°N | 71.56°W | Lago de Maracaibo | Brackish lake with surface area 13 210 km$^2$ |
| 35.44°S | 139.17°E | Lake Alexandrina | Freshwater shallow lake with surface area 649 km$^2$ |





**Table 3: Locations of 27 verification lake sites, lake morphological parameters measured by SYKE and from ECMWF Tco1279 fields**

| № | Lake ID | Lake name | Coordinates, deg | | In-situ depth, m | | In-situ area, km² | Modelled mean depth, m | |
|---|---------|-----------|----------|-----------|------|---------|------|-------------|------|
| | | | Latitude | Longitude | Mean | Maximum | | Operational | New |
| 1 | 44111001 | Pielinen | 63.2705 | 29.6067 | 10.1 | 61.0 | 894.2 | 5.0 | 10.0 |
| 2 | 42721001 | Kallavesi | 62.7616 | 27.7826 | 9.7 | 75.0 | 316.1 | 17.0 | 7.0 |
| 3 | 42111001 | Haukivesi | 62.1083 | 28.3887 | 9.1 | 55.0 | 560.4 | 2.0 | 7.0 |
| 4 | 41121001 | Saimaa | 61.3377 | 28.1158 | 10.8 | 85.8 | 1377.0 | 2.0 | 18.0 |
| 5 | 146311001 | Paajarvi1 | 62.8638 | 24.7894 | 3.8 | 14.9 | 29.5 | 25.0 | 2.9 |
| 6 | 147311001 | Nilakka | 63.1146 | 26.5268 | 4.9 | 21.7 | 169.0 | 17.0 | 7.0 |
| 7 | 147111001 | Konnevesi | 62.6326 | 26.6046 | 10.6 | 57.1 | 189.2 | 25.0 | 7.0 |
| 8 | 148211001 | Jaasjarvi | 61.6310 | 26.1351 | 4.6 | 28.2 | 81.1 | 25.0 | 6.1 |
| 9 | 142211001 | Paijanne | 61.6139 | 25.4820 | 14.1 | 86.0 | 864.9 | 13.0 | 13.9 |
| 10 | 141711001 | Ala-Rievel | 61.3035 | 26.1718 | 11.2 | 46.9 | 13.0 | 25.0 | 7.0 |
| 11 | 149321001 | Kyyvesi | 61.9988 | 27.0796 | 4.4 | 35.3 | 130.0 | 25.0 | 5.0 |
| 12 | 210821001 | Tuusulanja | 60.4414 | 25.0544 | 3.2 | 9.8 | 5.9 | 3.0 | 6.2 |
| 13 | 340311001 | Pyhajarvi | 61.0011 | 22.2913 | 5.5 | 26.2 | 155.2 | 4.5 | 5.0 |
| 14 | 357211001 | Langelmave | 61.5353 | 24.3705 | 6.8 | 59.3 | 133.0 | 25.0 | 7.0 |
| 15 | 358331003 | Paajarvi2 | 61.0635 | 25.1325 | 14.8 | 85.0 | 13.4 | 13.2 | 12.9 |
| 16 | 354121001 | Vaskivesi | 62.1416 | 23.7635 | 7.0 | 62.0 | 46.1 | 28.0 | 7.0 |
| 17 | 359311007 | Kuivajarvi | 60.7855 | 23.8596 | 2.2 | 9.9 | 8.2 | 9.0 | 7.0 |
| 18 | 353111001 | Nasijarvi | 61.6318 | 23.7505 | 14.7 | 65.6 | 210.6 | 28.0 | 7.0 |
| 19 | 470311001 | Lappajarvi | 63.1480 | 23.6706 | 6.9 | 36.0 | 145.5 | 2.0 | 18.0 |
| 20 | 595411001 | Pesiojarvi | 64.9451 | 28.6502 | 3.9 | 15.8 | 12.7 | 7.0 | 6.6 |
| 21 | 598111001 | Rehja-Nuas | 64.1840 | 28.0162 | 8.5 | 42.0 | 96.4 | 8.0 | 10.0 |
| 22 | 593111001 | Oulujarvi | 64.4500 | 26.9700 | 7.0 | 35.0 | 887.1 | 2.0 | 7.0 |
| 23 | 656321001 | Ounasjarvi | 68.3771 | 23.6016 | 6.6 | 31.0 | 6.9 | 3.0 | 7.0 |
| 24 | 655921001 | Unari | 67.1725 | 25.7112 | 7.0 | 24.8 | 29.1 | 2.0 | 7.0 |
| 25 | 676401001 | Kilpisjarv | 69.0070 | 20.8160 | 19.5 | 57.0 | 37.3 | 18.0 | 17.8 |
| 26 | 680721002 | Kevojarvi | 69.7515 | 27.0148 | 11.1 | 35.0 | 1.0 | 14.0 | 7.0 |
| 27 | 711111001 | Inarijarvi | 69.0821 | 27.9245 | 14.3 | 92.0 | 1039.4 | 14.0 | 14.0 |



**Table 4: Ice formation/disappearance dates for 2010-2014 of 27 verification sites; OBS – measured by SYKE, *GTZP_{OPR}* and *GTZL_{NEW}* – ECMWF experiments with operational and updated *D_water* respectively**

| № | Lake name | Year | 2010 | 2011 | 2012 | 2013 | 2014 | 2010 | 2011 | 2012 | 2013 | 2014 |
|---|---|---|---|---|---|---|---|---|---|---|---|---|
| | | Data | | | Ice melting | | | | | Ice freezing | | |
| 1 | Pielinen | OBS | 05/10 | 05/02 | 05/14 | 05/08 | 04/28 | 11/22 | 12/12 | 12/03 | 12/02 | 12/18 |
| | | *GTZP_{OPR}* | 05/23 | 05/26 | 05/21 | 05/24 | 05/17 | 11/11 | 11/21 | 11/10 | 11/25 | 11/08 |
| | | *GTZL_{NEW}* | 05/23 | 05/26 | 05/21 | 05/24 | 05/17 | 11/15 | 11/30 | 11/15 | 11/27 | 11/14 |
| 2 | Kallavesi | OBS | 05/08 | 05/06 | 05/08 | 05/06 | 04/19 | 11/29 | 99/99 | 01/01 | 12/09 | 12/24 |
| | | *GTZP_{OPR}* | 05/22 | 05/28 | 05/20 | 05/23 | 05/16 | 11/27 | 12/30 | 12/03 | 12/09 | 12/18 |
| | | *GTZL_{NEW}* | 05/22 | 05/26 | 05/21 | 05/24 | 05/18 | 11/15 | 11/30 | 11/30 | 11/30 | 11/22 |
| 3 | Haukivesi | OBS | 05/04 | 05/08 | 05/07 | 05/05 | 04/19 | 11/27 | 99/99 | **01/02** | 12/10 | **12/22** |
| | | *GTZP_{OPR}* | 05/21 | 05/25 | 05/18 | 05/21 | 05/13 | 11/08 | 11/18 | **10/29** | 11/24 | **10/23** |
| | | *GTZL_{NEW}* | 05/22 | 05/24 | 05/18 | 05/22 | 05/13 | 11/20 | 11/30 | **11/30** | 11/30 | **11/22** |
| 4 | Saimaa | OBS | 05/02 | 04/27 | 05/02 | 05/06 | 04/17 | 11/25 | **12/25** | **12/05** | *11/23* | **12/22** |
| | | *GTZP_{OPR}* | 05/15 | 05/18 | 05/11 | 05/15 | 04/28 | 11/18 | **11/21** | **10/31** | *11/26* | **10/23** |
| | | *GTZL_{NEW}* | 05/15 | 05/16 | 05/11 | 05/17 | 04/30 | 11/28 | **01/04** | **12/05** | *12/15* | **12/23** |
| 5 | Paajarvi1 | OBS | 05/02 | 04/27 | 05/03 | 05/02 | 04/18 | *11/20* | *12/12* | *11/30* | *11/29* | *12/05* |
| | | *GTZP_{OPR}* | 05/14 | 05/10 | 05/06 | 05/05 | 04/15 | *11/21* | *12/28* | *12/02* | *12/03* | *12/15* |
| | | *GTZL_{NEW}* | 05/13 | 05/10 | 05/06 | 05/05 | 04/15 | *10/16* | *11/16* | *10/27* | *10/20* | *10/22* |
| 6 | Nilakka | OBS | 05/05 | 05/02 | 05/08 | 05/06 | 04/23 | 11/17 | 12/12 | *11/29* | 11/23 | *12/02* |
| | | *GTZP_{OPR}* | 05/23 | 05/28 | 05/25 | 05/24 | 05/18 | 11/27 | 12/29 | *12/03* | 12/05 | *12/18* |
| | | *GTZL_{NEW}* | 05/23 | 05/26 | 05/25 | 05/24 | 05/17 | 11/12 | 11/30 | *11/17* | 11/26 | *11/20* |
| 7 | Konnevesi | OBS | 05/05 | 05/02 | 05/05 | 05/05 | 04/20 | 11/22 | *12/31* | *12/02* | 99/99 | *01/12* |
| | | *GTZP_{OPR}* | 05/08 | 05/09 | 05/03 | 05/04 | 04/14 | 11/22 | *01/01* | *12/03* | 12/08 | *12/22* |
| | | *GTZL_{NEW}* | 05/08 | 05/09 | 05/03 | 05/04 | 04/14 | 11/11 | *11/30* | *11/10* | 11/26 | *10/24* |
| 8 | Jaasjarvi | OBS | 04/27 | 04/27 | 05/01 | 05/02 | 04/11 | 11/21 | 99/99 | 01/01 | 12/05 | *12/19* |
| | | *GTZP_{OPR}* | 05/02 | 05/04 | 04/29 | 04/30 | 04/13 | 11/25 | 01/07 | 12/04 | 12/10 | *12/24* |
| | | *GTZL_{NEW}* | 05/02 | 05/04 | 04/29 | 05/01 | 04/13 | 11/18 | 12/08 | 11/30 | 11/27 | *10/24* |
| 9 | Paijanne | OBS | 05/04 | 04/30 | 05/01 | 05/03 | 04/12 | 11/27 | 99/99 | 01/01 | 12/15 | 12/26 |
| | | *GTZP_{OPR}* | 05/19 | 05/23 | 05/15 | 05/19 | 05/06 | 11/26 | 12/31 | 12/03 | 12/08 | 12/18 |
| | | *GTZL_{NEW}* | 05/19 | 05/23 | 05/17 | 05/19 | 05/05 | 11/27 | 12/31 | 12/03 | 12/10 | 12/19 |
| 10 | Ala-Rievel | OBS | 04/28 | 05/01 | 05/02 | 05/01 | 04/13 | 11/23 | 99/99 | 01/01 | 12/08 | *12/24* |
| | | *GTZP_{OPR}* | 05/02 | 05/03 | 04/28 | 05/01 | 04/13 | 11/26 | 01/08 | 12/04 | 12/10 | *12/23* |
| | | *GTZL_{NEW}* | 05/02 | 05/04 | 04/28 | 05/01 | 04/13 | 11/19 | 12/10 | 11/30 | 11/30 | *11/23* |
| 11 | Kyyvesi | OBS | 99/99 | 04/30 | 99/99 | 05/02 | 04/12 | *11/21* | 99/99 | *12/03* | **11/30** | *12/22* |
| | | *GTZP_{OPR}* | 05/09 | 05/11 | 05/05 | 05/05 | 04/15 | *11/25* | 01/03 | *12/04* | **12/10** | *12/19* |





| | | | | | | | | | | | | |
|---|---|---|---|---|---|---|---|---|---|---|---|---|
| | | *GTZL$_{NEW}$* | 05/10 | 05/11 | 05/05 | 05/06 | 04/15 | *11/10* | 11/21 | *11/09* | **11/26** | *10/23* |
| 12 | Tuusulanja | OBS | 04/21 | 04/25 | 04/24 | 04/29 | 04/05 | 11/09 | **11/21** | ***11/10*** | 11/25 | **12/01** |
| | | *GTZP$_{OPR}$* | 04/25 | 04/27 | 04/20 | 04/26 | 03/25 | 11/19 | **12/09** | *10/28* | 11/26 | **10/23** |
| | | *GTZL$_{NEW}$* | 04/25 | 04/27 | 04/20 | 04/26 | 03/23 | 11/20 | **01/02** | ***12/01*** | 11/30 | **12/02** |
| 13 | Pyhajarvi | OBS | 04/26 | 05/02 | 04/26 | 05/01 | 04/03 | 11/22 | 99/99 | 01/07 | 12/11 | **12/24** |
| | | *GTZP$_{OPR}$* | 05/04 | 05/09 | 04/26 | 05/04 | 04/07 | 11/18 | 12/09 | 12/01 | 11/29 | **12/04** |
| | | *GTZL$_{NEW}$* | 05/04 | 05/09 | 04/26 | 05/04 | 04/07 | 11/19 | 12/09 | 12/01 | 11/30 | **12/21** |
| 14 | Langelmave | OBS | 05/01 | 04/30 | 05/01 | 05/03 | 04/20 | 11/22 | 99/99 | 01/01 | **12/09** | *12/23* |
| | | *GTZP$_{OPR}$* | 05/04 | 05/08 | 04/29 | 05/02 | 04/13 | 11/27 | 01/09 | 12/05 | **01/13** | *12/25* |
| | | *GTZL$_{NEW}$* | 05/05 | 05/09 | 04/29 | 05/03 | 04/13 | 11/19 | 12/11 | 11/30 | **11/30** | *12/02* |
| 15 | Paajarvi2 | OBS | 99/99 | 99/99 | 99/99 | 99/99 | 99/99 | 99/99 | 99/99 | 99/99 | 99/99 | 99/99 |
| | | *GTZP$_{OPR}$* | 05/01 | 05/02 | 04/27 | 04/30 | 04/12 | 11/22 | 01/02 | 12/01 | 12/02 | 12/21 |
| | | *GTZL$_{NEW}$* | 05/01 | 05/02 | 04/27 | 04/30 | 04/12 | 11/22 | 01/02 | 12/01 | 12/02 | 12/19 |
| 16 | Vaskivesi | OBS | 04/27 | 04/28 | 04/30 | 05/01 | 04/12 | 11/23 | *12/31* | *12/02* | *12/08* | **11/30** |
| | | *GTZP$_{OPR}$* | 05/06 | 05/06 | 04/30 | 05/02 | 04/13 | 11/23 | *01/07* | *12/04* | *12/10* | **12/22** |
| | | *GTZL$_{NEW}$* | 05/05 | 05/08 | 04/30 | 05/03 | 04/13 | 11/13 | *12/07* | *11/10* | *11/26* | **11/22** |
| 17 | Kuivajarvi | OBS | 04/21 | 04/26 | 04/24 | 04/29 | 04/02 | 11/22 | 99/99 | 01/01 | 12/01 | *12/22* |
| | | *GTZP$_{OPR}$* | 05/01 | 05/05 | 04/26 | 05/01 | 04/09 | 11/21 | 01/01 | 12/01 | 12/01 | *12/21* |
| | | *GTZL$_{NEW}$* | 05/01 | 05/06 | 04/26 | 05/01 | 04/09 | 11/20 | 12/12 | 12/01 | 11/30 | *12/04* |
| 18 | Nasijarvi | OBS | 04/29 | 05/01 | 05/01 | 05/03 | 04/13 | *11/29* | 99/99 | *01/10* | 99/99 | *01/14* |
| | | *GTZP$_{OPR}$* | 05/07 | 05/10 | 04/29 | 05/04 | 04/13 | *11/28* | 01/10 | *12/08* | 01/14 | *12/26* |
| | | *GTZL$_{NEW}$* | 05/07 | 05/11 | 05/01 | 05/05 | 04/13 | *11/19* | 12/12 | *12/01* | 11/30 | *12/04* |
| 19 | Lappajarvi | OBS | 05/04 | 05/03 | 05/02 | 05/03 | 04/17 | **11/22** | **12/31** | **12/03** | **11/22** | **12/22** |
| | | *GTZP$_{OPR}$* | 05/16 | 05/13 | 05/09 | 05/09 | 04/18 | **10/17** | **11/16** | **10/27** | **10/19** | **10/22** |
| | | *GTZL$_{NEW}$* | 05/16 | 05/13 | 05/09 | 05/09 | 04/17 | **11/21** | **12/28** | **12/02** | **11/30** | **12/14** |
| 20 | Pesiojarvi | OBS | 99/99 | 99/99 | 99/99 | 99/99 | 99/99 | 99/99 | 99/99 | 99/99 | 99/99 | 99/99 |
| | | *GTZP$_{OPR}$* | 05/18 | 05/17 | 05/16 | 05/14 | 05/11 | 10/28 | 11/16 | 10/27 | 10/20 | 10/18 |
| | | *GTZL$_{NEW}$* | 05/18 | 05/17 | 05/16 | 05/14 | 05/11 | 10/27 | 11/16 | 10/26 | 10/20 | 10/18 |
| 21 | Rehja-Nuas | OBS | 05/04 | 05/02 | 05/01 | 05/04 | 04/20 | 11/14 | 99/99 | 11/09 | 11/27 | 11/07 |
| | | *GTZP$_{OPR}$* | 05/17 | 05/16 | 05/14 | 05/11 | 04/29 | 11/09 | 11/20 | 10/30 | 11/21 | 10/22 |
| | | *GTZL$_{NEW}$* | 05/17 | 05/16 | 05/14 | 05/11 | 04/28 | 11/10 | 11/21 | 11/08 | 11/22 | 10/23 |
| 22 | Oulujarvi | OBS | 05/15 | 05/10 | 05/13 | 05/14 | 05/02 | **11/24** | 99/99 | 99/99 | **12/04** | **12/18** |
| | | *GTZP$_{OPR}$* | 05/25 | 05/28 | 05/25 | 05/27 | 05/20 | **10/27** | 11/15 | 10/27 | **10/20** | **10/18** |
| | | *GTZL$_{NEW}$* | 05/25 | 05/28 | 05/25 | 05/27 | 05/20 | **11/11** | 11/21 | 11/10 | **11/15** | **11/08** |
| 23 | Ounasjarvi | OBS | 05/24 | 05/23 | 05/27 | 05/25 | 06/01 | 11/06 | **11/16** | 10/28 | 11/12 | 10/31 |
| | | *GTZP$_{OPR}$* | 06/03 | 06/02 | 05/31 | 05/30 | 06/05 | 10/15 | **10/14** | 10/18 | 10/16 | 10/13 |



| | | | | | | | | | | | | |
|---|---|---|---|---|---|---|---|---|---|---|---|---|
| | | *GTZL_NEW* | 06/03 | 06/02 | 05/31 | 05/30 | 06/05 | 10/25 | **11/11** | 10/21 | 10/19 | 10/17 |
| 24 | Unari | OBS | 05/19 | 05/15 | 05/21 | 05/18 | 05/23 | **11/06** | **11/19** | **10/26** | **11/08** | **11/02** |
| | | *GTZP_OPR* | 06/02 | 05/30 | 05/28 | 05/26 | 05/29 | **10/15** | **10/15** | **10/18** | **10/16** | **10/13** |
| | | *GTZL_NEW* | 06/02 | 05/30 | 05/28 | 05/26 | 05/28 | **10/30** | **11/15** | **10/23** | **10/20** | **10/17** |
| 25 | Kilpisjarv | OBS | 06/15 | 06/09 | 06/19 | 06/03 | 06/19 | 11/10 | 12/07 | 11/14 | 11/19 | 11/05 |
| | | *GTZP_OPR* | 06/24 | 06/13 | 06/24 | 06/05 | 06/24 | 10/25 | 11/10 | 10/23 | 10/21 | 10/22 |
| | | *GTZL_NEW* | 06/24 | 06/13 | 06/21 | 06/05 | 06/24 | 10/25 | 11/10 | 10/23 | 10/21 | 10/22 |
| 26 | Kevojarvi | OBS | 05/23 | 05/25 | 05/26 | 05/19 | 05/27 | 10/29 | 11/18 | 10/27 | 11/07 | 10/25 |
| | | *GTZP_OPR* | 06/07 | 06/07 | 06/01 | 06/01 | 06/08 | 10/30 | 11/16 | 10/28 | 10/23 | 10/19 |
| | | *GTZL_NEW* | 06/07 | 06/07 | 06/01 | 06/01 | 06/08 | 10/26 | 11/04 | 10/23 | 10/18 | 10/17 |
| 27 | Inarijarvi | OBS | 06/03 | 06/03 | 05/31 | 05/25 | 06/02 | 11/26 | 99/99 | 11/26 | 11/27 | 11/13 |
| | | *GTZP_OPR* | 06/09 | 06/08 | 06/05 | 06/02 | 06/07 | 11/07 | 11/21 | 11/10 | 11/09 | 11/02 |
| | | *GTZL_NEW* | 06/09 | 06/08 | 06/05 | 06/02 | 06/07 | 11/07 | 11/21 | 11/10 | 11/09 | 11/02 |





**Table 5: Locations of in-situ water surface temperature and ice formation/disappearance measurement points and distance between them for 27 verification sites; latitude and longitude in degrees, distance in km**

| Lake name | Measurement location (ML) coordinates, deg | | | | | | Distance between WST and ice ML, km |
|---|---|---|---|---|---|---|---|
| | Water surface temperature (WST) | | | Ice | | | |
| | Latitude | Longitude | Note | Latitude | Longitude | Note | |
| Pielinen | 63.2705 | 29.6067 | middle of the lake | 63.5418 | 29.1314 | on/close to the lake shore | 38.45 |
| Kallavesi | 62.7616 | 27.7826 | middle of the lake | 62.8993 | 27.7317 | on/close to the lake shore | 15.56 |
| Haukivesi | 62.1083 | 28.3887 | middle of the lake | 62.1107 | 28.6064 | on/close to the lake shore | 11.37 |
| Saimaa | 61.3377 | 28.1158 | middle of the lake | 61.5008 | 27.2636 | on/close to the lake shore | 49.00 |
| Paajarvi1 | 62.8638 | 24.7894 | middle of the lake | 62.8474 | 24.8142 | close to the lake shore | 2.22 |
| Nilakka | 63.1146 | 26.5268 | middle of the lake | 63.1993 | 26.6696 | on/close to the lake shore | 11.87 |
| Konnevesi | 62.6326 | 26.6046 | middle of the lake | 62.6166 | 26.3492 | on/close to the lake shore | 13.23 |
| Jaasjarvi | 61.6310 | 26.1351 | middle of the lake | 61.5674 | 26.0467 | on/close to the lake shore | 8.50 |
| Paijanne | 61.6139 | 25.4820 | middle of the lake | 61.1760 | 25.5362 | on/close to the lake shore | 48.88 |
| Ala-Rievel | 61.3035 | 26.1718 | middle of the lake | 61.3358 | 26.2012 | on/close to the lake shore | 3.93 |
| Kyyvesi | 61.9988 | 27.0796 | middle of the lake | 62.0127 | 27.1895 | on/close to the lake shore | 5.96 |
| Tuusulanja | 60.4414 | 25.0544 | middle of the lake | 60.4168 | 25.0427 | on/close to the lake shore | 2.82 |
| Pyhajarvi | 61.0011 | 22.2913 | middle of the lake | 61.1015 | 22.1802 | on/close to the lake shore | 12.70 |
| Langelmave | 61.5353 | 24.3705 | middle of the lake | 61.4180 | 24.1474 | on/close to the lake shore | 17.67 |
| Paajarvi2 | 61.0635 | 25.1325 | middle of the lake | | | no station | |
| Vaskivesi | 62.1416 | 23.7635 | middle of the lake | 62.1175 | 23.9249 | on/close to the lake shore | 8.84 |
| Kuivajarvi | 60.7855 | 23.8596 | middle of the lake | 60.7821 | 23.8383 | on/close to the lake shore | 1.22 |
| Nasijarvi | 61.6318 | 23.7505 | middle of the lake | 61.5086 | 23.7725 | on/close to the lake shore | 13.78 |
| Lappajarvi | 63.1480 | 23.6706 | middle of the lake | 63.2595 | 23.6353 | on/close to the lake shore | 12.55 |
| Pesiojarvi | 64.9451 | 28.6502 | middle of the lake | | | no station | |
| Rehja-Nuas | 64.1840 | 28.0162 | middle of the lake | 64.1616 | 28.2441 | on/close to the lake shore | 11.36 |
| Oulujarvi | 64.4500 | 26.9700 | middle of the lake | 64.5509 | 26.8240 | on/close to the lake shore | 13.26 |
| Ounasjarvi | 68.3771 | 23.6016 | close to lake shore | 68.3975 | 23.7170 | on/close to the lake shore | 5.26 |
| Unari | 67.1725 | 25.7112 | middle of the lake | 67.1366 | 25.7416 | on/close to the lake shore | 4.22 |
| Kilpisjarv | 69.0070 | 20.8160 | middle of the lake | 69.0497 | 20.7881 | on/close to the lake shore | 4.90 |
| Kevojarvi | 69.7515 | 27.0148 | middle of the lake | 69.7566 | 27.0031 | on/close to the lake shore | 0.72 |
| Inarijarvi | 69.0821 | 27.9245 | middle of the lake | 68.9577 | 27.6942 | middle of the lake | 16.65 |