# Peer review of "Upgraded global mapping information for earth system modelling: an application to surface water depth at ECMWF"

_Hydrology and Earth System Sciences, 2019_

## Referee Comment (RC1) · Anonymous Referee #1 · 10 Jul 2019

Comments to the manuscript of Margarita Choulga et al. "Upgraded mapping information for earth system modeling: an application to surface water depth at ECMWF"

Topicality. The present paper is devoted to the highly topical theme of the interaction between atmosphere and the underlying surface. Namely, the study concerns methodology of accounting for the influence of inland water bodies on the local weather and climatic conditions. This problem was raised relatively recently, but very quickly became one of the main in improving the numerical weather prediction. Thus, the problem discussed in the article is important and relevant.

Abstract. Should be rewritten. First 7 lines contain information about previous results.

Moreover, the FLake scheme is described in detail in the Introduction. In my opinion, the abstract should contain a brief, but close to complete information on new results obtained in the current study. In addition, it seems that the use of acronyms in the abstract is not the best idea.

Introduction.

Line 20. Authors wrote: "... thaw area are rich in nutrients, which affect the CO2 budget (Walter et 20 al., 2006;..."

I didn't find any mention of CO2 in Walter et al, 2006. In the study cited, CO2 is found only in the list of references, and the article itself is devoted to methane emission. Indeed, both gases are greenhouse gases and are formed by the decomposition of organic matter. But the conditions of their formation differ radically. Carbon dioxide is formed in aerobic conditions and methane in the complete absence of dissolved oxygen in water. In addition, these greenhouse gases are formed as a result of the activity of completely different microorganisms. Maybe the authors meant that due to the abundance of organic matter in the lakes of Siberia lakes produce a large amount of greenhouse gases? Moreover, the term "nutrients" in respect to lakes usually is applying to so-called "biogenic" elements, such as phosphorus and nitrogen. Suitable corrections are needed.

**Data**

No comments, just a question concerning the fourth distinction between GLDBv3 and GLDBv1.What are the "analytical equations to define the lake mean depth from the lakes' area and boreal zones climate type"? How they were derived, how to look at them or where they can be found? Please, add few words.

Methods

3.2 Updates

Lines 5 - 10 and Fig. 4, second from the left plot.
The authors can hang me, but I couldn't find any lake in this plot - neither lake Moondarra nor lake area Machattie. Even at 400% image magnification. I trust to the authors and their respected Australian experts, but something should be done to improve the Figure. .

Page 11. Lines 15, 17 and 21. Please, clarify: "on Fig. 7" or " in Fig. 7"

Verification and discussion

Page 14. Lines 30 - 31. The third and the fourth seasons are marked identically - (iii)

Actually the freshwater lakes have five main seasons at least. The fifth is the period of winter lake cooling between the temperature of maximal density and start of ice formation. During this period cooling takes place under so-called inverse stable density stratification conditions. Corrections are needed.

**4.2 Model verification results**

All the authors' explanations of the large values of errors in dates of ice-on and ice-off have the right to exist. Nevertheless, I'd like to put their attention to such parameter as ice albedo. The point is as follows. During melting the value of ice albedo radically decreases, that leads to the essential increase of the ice melting rate. If the ice albedo in numerical experiments was constant for the whole period of calculations that can lead to large mistakes in dates of ice melting at least. My advice to the authors: Add a few phrases about it.

Discussion No comments.

Conclusion No comments

Data availability

Please change the site of FLake model link to http://www.lakemodel.net

Summary The manuscript can be published after minor revision.
Please also note the supplement to this comment: https://www.hydrol-earth-syst-sci-discuss.net/hess-2019-234/hess-2019-234-RC1supplement.pdf

---

## Referee Comment (RC2) · Anonymous Referee #1 · 26 Jul 2019

Upgraded global mapping information for earth system modelling: an application to surface water depth at ECMWF

For the editor and authors.

The reviewer is completely satisfied with the authors ' response to the reviewer's comments. The manuscript can be published in its current form.

---

## Author Comment (AC1) · 26 Jul 2019

**Comments to the manuscript of Margarita Choulga et al. "Upgraded mapping information for earth system #modelling: an application to surface water depth at ECMWF"**

Thank you for the positive evaluation and useful comments. Below you will find our detailed responses to your comments.

**Topicality. The present paper is devoted to the highly topical theme of the interaction between atmosphere #and the underlying surface. Namely, the study concerns methodology of accounting for the influence of #inland water bodies on the local**
weather and climatic conditions. This problem was raised relatively recently, #but very quickly became one of the main in improving the numerical weather prediction. Thus, the problem #discussed in the article is important and relevant.

**Abstract. Should be rewritten. First 7 lines contain information about previous results. Moreover, the FLake #scheme is described in detail in the Introduction. In my opinion, the abstract should contain a brief, but #close to complete information on new results obtained in the current study. In addition, it seems that the #use of acronyms in the abstract is not the best idea.**

We propose to rewrite the Abstract in a following way: "Water bodies influence local weather and climate, especially in lake-rich areas. The FLake (Fresh-water Lake model) parametrization is employed in the Integrated Forecast System (IFS) of the European Centre for Medium-range Weather Forecasts (ECMWF) model which is used operationally to produce global weather predictions. Lake depth and lake fraction are the main driving parameters in the FLake parametrization. The lake parameter fields for IFS should be global and realistic, because FLake runs over all the grid boxes, and then only lake-related results are used further. In this study new datasets and methods for generating lake fraction and lake depth fields for IFS are proposed. The data include the new version of the Global Lake Database (GLDBv3) which contains depth estimates for unstudied lakes based on a geological approach, the General Bathymetric Chart of the Oceans and the Global Surface Water Explorer dataset which contains information on the spatial and temporal variability of surface water. The first new method suggested is a two-step lake fraction calculation; the first step is at 1 km grid resolution and the second is at the resolution of other grids in the IFS system. The second new method involves the use of a novel algorithm for ocean and inland water separation. This new algorithm may be used by anyone in the environmental modelling community. To assess the impact of using these innovations, in-situ measurements of lake depth, lake water surface temperature and ice formation/disappearance dates for 27 lakes collected by the Finnish Environment Institute were used. A set of offline

experiments, driven by atmospheric forcing from the ECMWF ERA5 Reanalysis were carried out using the IFS HTESSEL land surface model. In terms of lake depth, the new dataset shows a much lower mean absolute error, bias and error standard deviation compared to the reference set-up. In terms of lake water surface temperature, the mean absolute error is reduced by 13.4 %, the bias by 12.5 % and the error standard deviation by 20.3 %. Seasonal verification of the mixed layer depth temperature and ice formation/disappearance dates revealed a cold bias in the meteorological forcing from ERA5. Spring, summer and autumn verification scores confirm an overall reduction in the surface water temperature errors. For winter, no statistically significant change in the ice formation/disappearance date errors was detected.".

**Introduction.**

**Line 20. Authors wrote: "... thaw area are rich in nutrients, which affect the CO2 budget (Walter et al., #2006;..."**

**I didn't find any mention of CO2 in Walter et al, 2006. In the study cited, CO2 is found only in the list of #references, and the article itself is devoted to methane emission. Indeed, both gases are greenhouse gases #and are formed by the decomposition of organic matter. But the conditions of their formation differ radically. #Carbon dioxide is formed in aerobic conditions and methane in the complete absence of dissolved oxygen in #water. In addition, these greenhouse gases are formed as a result of the activity of completely different #microorganisms. Maybe the authors meant that due to the abundance of organic matter in the lakes of #Siberia lakes produce a large amount of greenhouse gases? #Moreover, the term "nutrients" in respect to lakes usually is applying to so-called "biogenic" elements, such #as phosphorus and nitrogen. #Suitable corrections are needed.**

Corrected as follows: "Lakes can also influence global climate by affecting the carbon cycle through carbon dioxide (CO2) and methane (CH4) emissions (Tranvik et al., 2009, Stepanenko et al., 2016). Small shallow thermokarst lakes located at Boreal and

Arctic latitudes in the permafrost thaw area are rich in organic matter from permafrost eroding into anaerobic lake bottoms (Walter et al., 2006; Stepanenko et al., 2012), which affect the CH4 budget, being as large as CO2 budget for these lakes (Walter et al., 2007).".

**Data**

**No comments, just a question concerning the fourth distinction between GLDBv3 and GLDBv1.What are the #"analytical equations to define the lake mean depth from the lakes' area and boreal zones climate type"? #How they were derived, how to look at them or where they can be found? #Please, add few words.**

Analytical equations to define the lake mean depth from the lakes' area and boreal zones climate type were developed using the study of Kitaev in 1984. He considered geographical zones of tundra, northern taiga, middle taiga and mixed forest, and presented his results in generalized tables. To improve the accuracy and usability, the tables were transformed into analytical equations approximating statistical dependencies and are presented in Table 3 of Section 4.4 in Choulga et al., 2014. In the text for the fourth point we suggest adding the citation on Choulga et al., 2014.

**Methods**

**3.2 Updates**

**Lines 5 – 10 and Fig. 4, second from the left plot.**

**The authors can hang me, but I couldn't find any lake in this plot - neither lake Moondarra nor lake area #Machattie. Even at 400% image magnification. #I trust to the authors and their respected Australian experts, but something should be done to improve the #Figure.**

Yes, these lakes are practically not visible due to the scale. We have added red circles around these two lakes to specify their location.

**Page 11. Lines 15, 17 and 21. Please, clarify: "on Fig. 7" or " in Fig. 7".**

Definitely "in Fig. 7", thank you for pointing on that.

**Verification and discussion**

**Page 14. Lines 30 – 31. The third and the fourth seasons are marked identically – (iii)**

**Actually the freshwater lakes have five main seasons at least. The fifth is the period of winter lake cooling #between the temperature of maximal density and start of ice formation. During this period cooling takes #place under so-called inverse stable density stratification conditions. #Corrections are needed.**

Corrected.

**4.2 Model verification results**

**All the authors' explanations of the large values of errors in dates of ice-on and ice-off have the right to #exist. Nevertheless, I'd like to put their attention to such parameter as ice albedo. The point is as follows. #During melting the value of ice albedo radically decreases, that leads to the essential increase of the ice #melting rate. If the ice albedo in numerical experiments was constant for the whole period of calculations #that can lead to large mistakes in dates of ice melting at least. #My advice to the authors: Add a few phrases about it.**

In the IFS the following parameterization of the ice surface albedo $\alpha i$ with respect to solar radiation is adopted: $\alpha i = \alpha imax - (\alpha imax - \alpha imin) \cdot e - C\alpha$ (Tf − Tice) / Tf, where $\alpha imax = 0.7$ and $\alpha imin = 0.4$ are maximum and minimum values of the ice albedo, respectively, $C\alpha = 95.6$ is a fitting coefficient, Tf = 273.15 K is the fresh-water freezing point, and Tice is temperature at the ice upper surface. The presence of snow over lake ice and its seasonal changes are parametrized in the equation above as a function of Tice. During the melting season, the ice surface temperature is close to the fresh-water freezing point. The presence of wet snow, puddles, melt-water ponds is again parametrized implicitly and results in a decrease of the area-averaged surface

albedo. The water surface albedo with respect to solar radiation, $\alpha w = 0.07$, is assumed to be constant. For more information "IFS Documentation CY43R3 - Part IV: Physical processes" Section 8.8.3, ECMWF, 4, 2017, can be used. In the text we suggest adding this information to the model description (in Section 1): "The ice albedo is dependent on the temperature at the ice upper surface and is lower in spring, during the melting period, see (IFS Documentation, 2017) for more details.".

**Discussion #No comments.**

**Conclusion #No comments.**

**Data availability**

**Please change the site of FLake model link to http://www.lakemodel.net**

Corrected.

**Summary #The manuscript can be published after minor revision.**

Please also note the supplement to this comment:
https://www.hydrol-earth-syst-sci-discuss.net/hess-2019-234/hess-2019-234-AC1-supplement.zip

---

## Author Comment (AC2) · 29 Jul 2019

Dear Anonymous Reviewer #1, Thank you for your kind words.

Dear Editor, We are looking forward to proceed with publication. Thank you.

Sincerely, Margarita
* * *

---

## Referee Comment (RC3) · Anonymous Referee #2 · 4 Aug 2019

- The abstract could be reviewed to make it easier for any kind of readers to understand what is this work about and what is addressed and expected. The abstract here started with many previous results which make it not easy to understand it.

- The paper is well structured and all information references are well cited.

- A lot of data are engaged and comparisons with other models and validation are present. Obtaining accurate and timely lake surface water temperature analyses from remote sensing remains difficult. Data gaps, cloud contamination, variations in temperature atmospheric profiles and moisture, and a lack of in situ observations provide challenges for satellite-derived surface water temperature for climatological analysis or

input into geophysical models. The authors used different sources of data including Reanalysis to test the operational and new lake depths. The seasonal and annual variations may need further assessment mainly if the authors got time-series data. The upscaling or downscaling of satellite resolution is always a challenge but it is well addressed in this work.

- I congratulate the authors for such rich and rigorous paper, which will definitely add to the knowledge of the scientific committee in this field.

---

## Author Comment (AC3) · 8 Aug 2019

Thank you for the positive evaluation and useful comments. Below you will find our detailed responses to your comments.

- The abstract could be reviewed to make it easier for any kind of readers to understand what is this work about and what is addressed and expected. The abstract here started with many previous results which make it not easy to understand it.

[Figure]

We propose to rewrite the Abstract in a following way: "Water bodies influence local weather and climate, especially in lake-rich areas. The FLake (Fresh-water Lake model) parametrization is employed in the Integrated Forecast System (IFS) of the European Centre for Medium-range Weather Forecasts (ECMWF) model which is used operationally to produce global weather predictions. Lake depth and lake fraction are the main driving parameters in the FLake parametrization. The lake parameter fields for IFS should be global and realistic, because FLake runs over all the grid boxes, and then only lake-related results are used further. In this study new datasets and methods for generating lake fraction and lake depth fields for IFS are proposed. The data include the new version of the Global Lake Database (GLDBv3) which contains depth estimates for unstudied lakes based on a geological approach, the General Bathymetric Chart of the Oceans and the Global Surface Water Explorer dataset which contains information on the spatial and temporal variability of surface water. The first new method suggested is a two-step lake fraction calculation; the first step is at 1 km grid resolution and the second is at the resolution of other grids in the IFS system. The second new method involves the use of a novel algorithm for ocean and inland water separation. This new algorithm may be used by anyone in the environmental modelling community. To assess the impact of using these innovations, in-situ measurements of lake depth, lake water surface temperature and ice formation/disappearance dates for 27 lakes collected by the Finnish Environment Institute were used. A set of offline experiments, driven by atmospheric forcing from the ECMWF ERA5 Reanalysis were carried out using the IFS HTESSEL land surface model. In terms of lake depth, the new dataset shows a much lower mean absolute error, bias and error standard deviation compared to the reference set-up. In terms of lake water surface temperature, the mean absolute error is reduced by 13.4 %, the bias by 12.5 % and the error standard deviation by 20.3 %. Seasonal verification of the mixed layer depth temperature and ice formation/disappearance dates revealed a cold bias in the meteorological forcing from ERA5. Spring, summer and autumn verification scores confirm an overall reduction in the surface water temperature errors. For winter, no statistically significant change in

the ice formation/disappearance date errors was detected.".

- The paper is well structured and all information references are well cited.

- A lot of data are engaged and comparisons with other models and validation are present. Obtaining accurate and timely lake surface water temperature analyses from remote sensing remains difficult. Data gaps, cloud contamination, variations in temperature atmospheric profiles and moisture, and a lack of in situ observations provide challenges for satellite-derived surface water temperature for climatological analysis or input into geophysical models. The authors used different sources of data including Reanalysis to test the operational and new lake depths. The seasonal and annual variations may need further assessment mainly if the authors got time-series data. The upscaling or downscaling of satellite resolution is always a challenge but it is well addressed in this work.

Currently we are gathering satellite-based data of surface water temperature for several hundred lakes all over the globe to have a more detailed analysis of seasonal and annual lake surface water temperature variations and ice formation. We mention the importance of the remote sensing data for lakes in the Discussion section: ". . . it would be useful to compare model results with measurements from the other countries and climate zones as IFS is a global forecasting system. For that, data from remote sensing could be beneficial, although they contain gaps and cloud contamination problems.".

- I congratulate the authors for such rich and rigorous paper, which will definitely add to the knowledge of the scientific committee in this field.

Please also note the supplement to this comment:
https://www.hydrol-earth-syst-sci-discuss.net/hess-2019-234/hess-2019-234-AC3-supplement.zip

---

## Author Comment (AC4) · 17 Aug 2019

Final Author Comments to the Anonymous Reviewer #1 and Anonymous Reviewer #2 comments to the manuscript of Margarita Choulga et al. "Upgraded mapping information for earth system modelling: an application to surface water depth at ECMWF"

Dear Anonymous Reviewer #1 and Anonymous Reviewer #2, thank you for the positive evaluation and useful comments. Below you will find our detailed responses to your comments. Dear Editor in the supplement there is final version of our manuscript.

Anonymous Reviewer #1 comments and Authors reply

[Figure]

Abstract. Should be rewritten. First 7 lines contain information about previous results. Moreover, the FLake scheme is described in detail in the Introduction. In my opinion, the abstract should contain a brief, but close to complete information on new results obtained in the current study. In addition, it seems that the use of acronyms in the abstract is not the best idea.

We propose to rewrite the Abstract in a following way: "Water bodies influence local weather and climate, especially in lake-rich areas. The FLake (Fresh-water Lake model) parametrization is employed in the Integrated Forecast System (IFS) of the European Centre for Medium-range Weather Forecasts (ECMWF) model which is used operationally to produce global weather predictions. Lake depth and lake fraction are the main driving parameters in the FLake parametrization. The lake parameter fields for IFS should be global and realistic, because FLake runs over all the grid boxes, and then only lake-related results are used further. In this study new datasets and methods for generating lake fraction and lake depth fields for IFS are proposed. The data include the new version of the Global Lake Database (GLDBv3) which contains depth estimates for unstudied lakes based on a geological approach, the General Bathymetric Chart of the Oceans and the Global Surface Water Explorer dataset which contains information on the spatial and temporal variability of surface water. The first new method suggested is a two-step lake fraction calculation; the first step is at 1 km grid resolution and the second is at the resolution of other grids in the IFS system. The second new method involves the use of a novel algorithm for ocean and inland water separation. This new algorithm may be used by anyone in the environmental modelling community. To assess the impact of using these innovations, in-situ measurements of lake depth, lake water surface temperature and ice formation/disappearance dates for 27 lakes collected by the Finnish Environment Institute were used. A set of offline experiments, driven by atmospheric forcing from the ECMWF ERA5 Reanalysis were carried out using the IFS HTESSEL land surface model. In terms of lake depth, the new dataset shows a much lower mean absolute error, bias and error standard deviation compared to the reference set-up. In terms of lake water surface temperature, the

mean absolute error is reduced by 13.4 %, the bias by 12.5 % and the error standard deviation by 20.3 %. Seasonal verification of the mixed layer depth temperature and ice formation/disappearance dates revealed a cold bias in the meteorological forcing from ERA5. Spring, summer and autumn verification scores confirm an overall reduction in the surface water temperature errors. For winter, no statistically significant change in the ice formation/disappearance date errors was detected.".

Introduction. Line 20. Authors wrote: ". . . thaw area are rich in nutrients, which affect the CO2 budget (Walter et al., 2006;. . ." I didn't find any mention of CO2 in Walter et al, 2006. In the study cited, CO2 is found only in the list of references, and the article itself is devoted to methane emission. Indeed, both gases are greenhouse gases and are formed by the decomposition of organic matter. But the conditions of their formation differ radically. Carbon dioxide is formed in aerobic conditions and methane in the complete absence of dissolved oxygen in water. In addition, these greenhouse gases are formed as a result of the activity of completely different microorganisms. Maybe the authors meant that due to the abundance of organic matter in the lakes of Siberia lakes produce a large amount of greenhouse gases? Moreover, the term "nutrients" in respect to lakes usually is applying to so-called "biogenic" elements, such as phosphorus and nitrogen. Suitable corrections are needed.

Corrected as follows: "Lakes can also influence global climate by affecting the carbon cycle through carbon dioxide (CO2) and methane (CH4) emissions (Tranvik et al., 2009, Stepanenko et al., 2016). Small shallow thermokarst lakes located at Boreal and Arctic latitudes in the permafrost thaw area are rich in organic matter from permafrost eroding into anaerobic lake bottoms (Walter et al., 2006; Stepanenko et al., 2012), which affect the CH4 budget, being as large as CO2 budget for these lakes (Walter et al., 2007).".

Data. No comments, just a question concerning the fourth distinction between GLDBv3 and GLDBv1.What are the "analytical equations to define the lake mean depth from the lakes' area and boreal zones climate type"? How they were derived, how to look at

them or where they can be found? Please, add few words.

Analytical equations to define the lake mean depth from the lakes' area and boreal zones climate type were developed using the study of Kitaev in 1984. He considered geographical zones of tundra, northern taiga, middle taiga and mixed forest, and presented his results in generalized tables. To improve the accuracy and usability, the tables were transformed into analytical equations approximating statistical dependencies and are presented in Table 3 of Section 4.4 in Choulga et al., 2014. In the text for the fourth point we suggest adding the citation on Choulga et al., 2014.

Methods. 3.2 Updates. Lines 5 – 10 and Fig. 4, second from the left plot. The authors can hang me, but I couldn't find any lake in this plot - neither lake Moondarra nor lake area Machattie. Even at 400% image magnification. I trust to the authors and their respected Australian experts, but something should be done to improve the Figure.

Yes, these lakes are practically not visible due to the scale. We have added red circles around these two lakes to specify their location.

Methods. 3.2 Updates. Page 11. Lines 15, 17 and 21. Please, clarify: "on Fig. 7" or " in Fig. 7".

Definitely "in Fig. 7", thank you for pointing on that.

Verification and discussion. Page 14. Lines 30 – 31. The third and the fourth seasons are marked identically – (iii) Actually the freshwater lakes have five main seasons at least. The fifth is the period of winter lake cooling between the temperature of maximal density and start of ice formation. During this period cooling takes place under so-called inverse stable density stratification conditions. Corrections are needed.

Corrected.

4.2 Model verification results. All the authors' explanations of the large values of errors in dates of ice-on and ice-off have the right to exist. Nevertheless, I'd like to put their attention to such parameter as ice albedo. The point is as follows. During melting the

value of ice albedo radically decreases, that leads to the essential increase of the ice melting rate. If the ice albedo in numerical experiments was constant for the whole period of calculations that can lead to large mistakes in dates of ice melting at least. My advice to the authors: Add a few phrases about it.

In the IFS the following parameterization of the ice surface albedo $\alpha i$ with respect to solar radiation is adopted: $\alpha i = \alpha imax - (\alpha imax - \alpha imin) \cdot e{-}C\alpha$ (Tf $-$ Tice) / Tf, where $\alpha imax = 0.7$ and $\alpha imin = 0.4$ are maximum and minimum values of the ice albedo, respectively, $C\alpha = 95.6$ is a fitting coefficient, Tf = 273.15 K is the fresh-water freezing point, and Tice is temperature at the ice upper surface. The presence of snow over lake ice and its seasonal changes are parametrized in the equation above as a function of Tice. During the melting season, the ice surface temperature is close to the fresh-water freezing point. The presence of wet snow, puddles, melt-water ponds is again parametrized implicitly and results in a decrease of the area-averaged surface albedo. The water surface albedo with respect to solar radiation, $\alpha w = 0.07$, is assumed to be constant. For more information "IFS Documentation CY43R3 - Part IV: Physical processes" Section 8.8.3, ECMWF, 4, 2017, can be used. In the text we suggest adding this information to the model description (in Section 1): "The ice albedo is dependent on the temperature at the ice upper surface and is lower in spring, during the melting period, see (IFS Documentation, 2017) for more details.".

Data availability. Please change the site of FLake model link to http://www.lakemodel.net

Corrected.

Anonymous Reviewer #1 reply to the Author and Authors comment

The reviewer is completely satisfied with the authors' response to the reviewer's comments. The manuscript can be published in its current form.

Thank you for your kind words. We are looking forward to proceed with publication.
Anonymous Reviewer #2 comments and Authors reply

- The abstract could be reviewed to make it easier for any kind of readers to understand what is this work about and what is addressed and expected. The abstract here started with many previous results which make it not easy to understand it.

We propose to rewrite the Abstract in a following way: "Water bodies influence local weather and climate, especially in lake-rich areas. The FLake (Fresh-water Lake model) parametrization is employed in the Integrated Forecast System (IFS) of the European Centre for Medium-range Weather Forecasts (ECMWF) model which is used operationally to produce global weather predictions. Lake depth and lake fraction are the main driving parameters in the FLake parametrization. The lake parameter fields for IFS should be global and realistic, because FLake runs over all the grid boxes, and then only lake-related results are used further. In this study new datasets and methods for generating lake fraction and lake depth fields for IFS are proposed. The data include the new version of the Global Lake Database (GLDBv3) which contains depth estimates for unstudied lakes based on a geological approach, the General Bathymetric Chart of the Oceans and the Global Surface Water Explorer dataset which contains information on the spatial and temporal variability of surface water. The first new method suggested is a two-step lake fraction calculation; the first step is at 1 km grid resolution and the second is at the resolution of other grids in the IFS system. The second new method involves the use of a novel algorithm for ocean and inland water separation. This new algorithm may be used by anyone in the environmental modelling community. To assess the impact of using these innovations, in-situ measurements of lake depth, lake water surface temperature and ice formation/disappearance dates for 27 lakes collected by the Finnish Environment Institute were used. A set of offline experiments, driven by atmospheric forcing from the ECMWF ERA5 Reanalysis were carried out using the IFS HTESSEL land surface model. In terms of lake depth, the new dataset shows a much lower mean absolute error, bias and error standard deviation compared to the reference set-up. In terms of lake water surface temperature, the

mean absolute error is reduced by 13.4 %, the bias by 12.5 % and the error standard deviation by 20.3 %. Seasonal verification of the mixed layer depth temperature and ice formation/disappearance dates revealed a cold bias in the meteorological forcing from ERA5. Spring, summer and autumn verification scores confirm an overall reduction in the surface water temperature errors. For winter, no statistically significant change in the ice formation/disappearance date errors was detected.".

- A lot of data are engaged and comparisons with other models and validation are present. Obtaining accurate and timely lake surface water temperature analyses from remote sensing remains difficult. Data gaps, cloud contamination, variations in temperature atmospheric profiles and moisture, and a lack of in situ observations provide challenges for satellite-derived surface water temperature for climatological analysis or input into geophysical models. The authors used different sources of data including Reanalysis to test the operational and new lake depths. The seasonal and annual variations may need further assessment mainly if the authors got time-series data. The upscaling or downscaling of satellite resolution is always a challenge but it is well addressed in this work.

Currently we are gathering satellite-based data of surface water temperature for several hundred lakes all over the globe to have a more detailed analysis of seasonal and annual lake surface water temperature variations and ice formation. We mention the importance of the remote sensing data for lakes in the Discussion section: "... it would be useful to compare model results with measurements from the other countries and climate zones as IFS is a global forecasting system. For that, data from remote sensing could be beneficial, although they contain gaps and cloud contamination problems.".

Please also note the supplement to this comment:
https://www.hydrol-earth-syst-sci-discuss.net/hess-2019-234/hess-2019-234-AC4-supplement.zip